# Proper Value Equivalence

**Christopher Grimm**
Computer Science & Engineering
University of Michigan
crgrimm@umich.edu

**André Barreto, Gregory Farquhar,**
**David Silver, Satinder Singh**
DeepMind
{andrebarreto,gregfar,
davidsilver,baveja}@google.com

## Abstract

One of the main challenges in model-based reinforcement learning (RL) is to decide which aspects of the environment should be modeled. The value-equivalence (VE) principle proposes a simple answer to this question: a model should capture the aspects of the environment that are relevant for value-based planning. Technically, VE distinguishes models based on a set of policies and a set of functions: a model is said to be VE to the environment if the Bellman operators it induces for the policies yield the correct result when applied to the functions. As the number of policies and functions increase, the set of VE models shrinks, eventually collapsing to a single point corresponding to a perfect model. A fundamental question underlying the VE principle is thus how to select the smallest sets of policies and functions that are sufficient for planning. In this paper we take an important step towards answering this question. We start by generalizing the concept of VE to order-$k$ counterparts defined with respect to $k$ applications of the Bellman operator. This leads to a family of VE classes that increase in size as $k \to \infty$. In the limit, all functions become value functions, and we have a special instantiation of VE which we call proper VE or simply PVE. Unlike VE, the PVE class may contain multiple models even in the limit when all value functions are used. Crucially, all these models are sufficient for planning, meaning that they will yield an optimal policy despite the fact that they may ignore many aspects of the environment. We construct a loss function for learning PVE models and argue that popular algorithms such as MuZero can be understood as minimizing an upper bound for this loss. We leverage this connection to propose a modification to MuZero and show that it can lead to improved performance in practice.

## 1 Introduction

It has long been argued that, in order for reinforcement learning (RL) agents to solve truly complex tasks, they must build a model of the environment that allows for counterfactual reasoning [29]. Since representing the world in all its complexity is a hopeless endeavor, especially under capacity constraints, the agent must be able to ignore aspects of the environment that are irrelevant for its purposes. This is the premise behind the *value equivalence* (VE) principle, which provides a formalism for focusing on the aspects of the environment that are crucial for value-based planning [17].

VE distinguishes models based on a set of policies and a set of real-valued scalar functions of state (henceforth, just functions). Roughly, a model is said to be VE to the environment if the Bellman operators it induces for the policies yield the same result as the environment's Bellman operators when applied to the functions. The policies and functions thus become a "language" to specify which parts of the environment a model should capture. As the number of policies and functions increase the requirements on the model become more stringent, which is to say that the class of VE models shrinks. In the limit, the VE class collapses to a single point corresponding to a perfect model.

35th Conference on Neural Information Processing Systems (NeurIPS 2021).

Although this result is reassuring, in practice we want to stop short of collapsing—after all, at this point the agent is no longer ignoring irrelevant aspects of the environment.

A fundamental question is thus how to select the smallest sets of policies and functions such that a resulting VE model is sufficient for planning. In this paper we take an important additional step in this direction: we show that the VE principle can be formulated with respect to *value functions* only. This result drastically reduces the space of functions that must be considered by VE, as in general only a small fraction of the set of all functions will qualify as value functions in a given environment. Since every policy has an associated value function, this new formulation of VE removes the need for selecting functions, only requiring policies. We name our new formulation *proper value equivalence* (PVE) to emphasize its explicit use of value functions.

PVE has several desirable properties. Unlike with VE, the class of PVE models does not collapse to a singleton in the limit. This means that, even if *all* value functions are used, we generally end up with multiple PVE models—which can be beneficial if some of these are easier to learn or represent than others. Crucially, all of these models are sufficient for planning, meaning that *they will yield an optimal policy despite the fact that they may ignore many aspects of the environment*.

Finally, we make more precise Grimm et al.'s [17] suggestion that the VE principle may help explain the good empirical performance of several modern algorithms [38, 33, 24, 12, 30]. Specifically, we show that, with mild assumptions, minimizing the loss of the MuZero algorithm [31] can be understood as minimizing a PVE error. We then leverage this connection to suggest a modification to MuZero and show a small but significant improvement in the Atari Learning Environment [3].

## 2 Background

The agent's interaction with the environment will be modeled as a *Markov decision process* (MDP) $\mathcal{M} \equiv \langle \mathcal{S}, \mathcal{A}, r, p, \gamma \rangle$, where $\mathcal{S}$ and $\mathcal{A}$ are the state and action spaces, $r(s, a)$ is the expected reward following taking $a$ from $s$, $p(s'|s, a)$ is the transition kernel and $\gamma \in [0, 1)$ is a discount factor [27]. A *policy* is a mapping $\pi : \mathcal{S} \mapsto \mathcal{P}(\mathcal{A})$, where $\mathcal{P}(\mathcal{A})$ is the space of probability distributions over $\mathcal{A}$; we define $\mathbb{\Pi} \equiv \{\pi \,|\, \pi : \mathcal{S} \mapsto \mathcal{P}(\mathcal{A})\}$ as the set of all possible policies. A policy $\pi$ is *deterministic* if $\pi(a|s) > 0$ for only one action $a$ per state $s$. A policy's *value function* is defined as

$$v_\pi(s) \equiv \mathbb{E}_\pi\Big[ \sum_{i=0}^{\infty} \gamma^i r(S_{t+i}, A_{t+i}) \,|\, S_t = s \Big], \tag{1}$$

where $\mathbb{E}_\pi[\cdot]$ denotes expectation over the trajectories induced by $\pi$ and the random variables $S_t$ and $A_t$ indicate the state occupied and the action selected by the agent at time step $t$.

The agent's goal is to find a policy $\pi \in \mathbb{\Pi}$ that maximizes the value of every state [36, 37]. Usually, a crucial step to carry out this search is to compute the value function of candidate policies. This process can be cast in terms of the policy's *Bellman operator*:

$$\mathcal{T}_\pi[v](s) \equiv \mathbb{E}_{A \sim \pi(\cdot|s), S' \sim p(\cdot|s, A)} \left[ r(s, A) + \gamma v(S') \right], \tag{2}$$

where $v$ is any function in the space $\mathbb{V} \equiv \{f \,|\, f : \mathcal{S} \mapsto \mathbb{R}\}$. It is known that $\lim_{n \to \infty} \mathcal{T}_\pi^n v = v_\pi$, that is, starting from any $v \in \mathbb{V}$, the repeated application of $\mathcal{T}_\pi$ will eventually converge to $v_\pi$. Since in RL the agent does not know $p$ and $r$, it cannot apply (2) directly. One solution is to learn a *model* $\tilde{m} \equiv (\tilde{r}, \tilde{p})$ and use it to compute (2) with $p$ and $r$ replaced by $\tilde{p}$ and $\tilde{r}$ [36]. We denote the set of all models as $\mathbb{M}$.

The *value equivalence principle* defines a model as value equivalent (VE) to the environment $m^* \equiv (r, p)$ with respect to a set of policies $\Pi$ and a set of functions $\mathcal{V}$ if it produces the same Bellman updates as $m^*$ when using $\Pi$ and $\mathcal{V}$ [17]. Classes of such models are expressed as follows:

$$\mathcal{M}(\Pi, \mathcal{V}) \equiv \{\tilde{m} \in \mathcal{M} : \tilde{\mathcal{T}}_\pi v = \mathcal{T}_\pi v \; \forall \pi \in \Pi, v \in \mathcal{V}\} \tag{3}$$

where $\mathcal{M} \subseteq \mathbb{M}$ is a class of models, $\tilde{\mathcal{T}}_\pi$ denotes one application of the Bellman operator induced by model $\tilde{m}$ and policy $\pi$ to function $v$, and $\mathcal{T}_\pi$ is environment's Bellman operator for $\pi$.

Grimm et al. [17] showed that the VE principle can be used to learn models that disregard aspects of the environment which are not related to the task of interest.[1] Classical approaches to model learning

---

[1] A related approach is taken in value-aware model learning [11] which minimizes the discrepancy between the Bellman optimality operators induced by the model and the environment.

do not take the eventual use of the model into account, potentially modeling irrelevant aspects of the environment. Accordingly, Grimm et al. [17] have shown that, under the same capacity constraints, models learned using VE can outperform their classical counterparts.

# 3 Proper value equivalence

One can define a spectrum of VE classes corresponding to different numbers of applications of the Bellman operator. We define an order-$k$ VE class as:

$$\mathcal{M}^k(\Pi, \mathcal{V}) \equiv \{\tilde{m} \in \mathcal{M} : \tilde{\mathcal{T}}_\pi^k v = \mathcal{T}_\pi^k v \ \forall \pi \in \Pi, v \in \mathcal{V}\} \tag{4}$$

where $\tilde{\mathcal{T}}_\pi^k v$ denotes $k$ applications of $\tilde{\mathcal{T}}_\pi$ to $v$. Under our generalized definition of VE, Grimm et al. [17] studied order-one VE classes of the form $\mathcal{M}^1(\Pi, \mathcal{V})$. They have shown that $\mathcal{M}^1(\mathbb{\Pi}, \mathbb{V})$ either contains only the environment or is empty. This is not generally true for $k > 1$. The limiting behavior of order-$k$ value equivalent classes can be described as follows

**Proposition 1.** *Let $\mathcal{V}$ be a set of functions such that if $v \in \mathcal{V}$ then $\mathcal{T}_\pi v \in \mathcal{V}$ for all $\pi \in \Pi$. Then, for $k, K \in \mathbb{Z}^+$ such that $k$ divides $K$, it follows that:*

*(i) For any $\mathcal{M} \subseteq \mathbb{M}$ and any $\Pi \subseteq \mathbb{\Pi}$, we have that $\mathcal{M}^k(\Pi, \mathcal{V}) \subseteq \mathcal{M}^K(\Pi, \mathcal{V})$.*

*(ii) If $\Pi$ is non-empty and $\mathcal{V}$ contains at least one constant function, then there exist environments such that $\mathbb{M}^k(\Pi, \mathcal{V}) \subset \mathbb{M}^K(\Pi, \mathcal{V})$.*

We defer all proofs of theoretical results to Appendix A.2. Based on Proposition 1 we can relate different VE model classes according to the greatest common divisor of their respective orders; specifically, two classes $\mathcal{M}^k(\Pi, \mathcal{V})$ and $\mathcal{M}^K(\Pi, \mathcal{V})$ will intersect at $\mathcal{M}^{\gcd(k,K)}(\Pi, \mathcal{V})$ (Figure 1). Proposition 1 also implies that, in contrast to order-one VE classes, higher order VE classes potentially include multiple models, even if VE is defined with respect to all policies $\mathbb{\Pi}$ and all functions $\mathbb{V}$. In addition, the size of a VE class cannot decrease as we increase its order from $k$ to a multiple of $k$ (and in some cases it will strictly increase). This invites the question of what happens in the limit as we keep increasing the VE order. To answer this question, we introduce a crucial concept for this paper:

**Definition 1.** *(Proper value equivalence). Given a set of policies $\Pi \subseteq \mathbb{\Pi}$, let*

$$\mathcal{M}^\infty(\Pi) = \lim_{k \to \infty} \mathcal{M}^k(\Pi, \mathbb{V}) = \{\tilde{m} \in \mathcal{M} : \tilde{v}_\pi = v_\pi \ \forall \pi \in \Pi\}, \tag{5}$$

*where $\tilde{v}_\pi$ and $v_\pi$ are the value functions of $\pi$ induced by model $\tilde{m}$ and the environment. We say that each $\tilde{m} \in \mathcal{M}^\infty(\Pi)$ is a **proper value equivalent** model to the environment with respect to $\Pi$.*

Because the process of repeatedly applying a policy's Bellman operator to a function converges to the same fixed point regardless of the function, in an order-$\infty$ VE class the set $\Pi$ uniquely determines the set $\mathcal{V}$. This reduces the problem of defining $\Pi$ and $\mathcal{V}$ to defining the former only. Also, since all functions in an order-$\infty$ VE are *value* functions, we call it *proper VE* or PVE.

It is easy to show that Proposition 1 is valid for any $k \in \mathbb{Z}^+$ when $K = \infty$ (Corollary 2 in Appendix A.2). Thus, in some sense, $\mathcal{M}^\infty$ is the "biggest" VE class. It is also possible to define this special VE class in terms of any other:

**Proposition 2.** *For any $\Pi \subseteq \mathbb{\Pi}$ and any $k \in \mathbb{Z}^+$ it follows that*

$$\mathcal{M}^\infty(\Pi) = \bigcap_{\pi \in \Pi} \mathcal{M}^k(\{\pi\}, \{v_\pi\}), \tag{6}$$

where $v_\pi$ is the value of policy $\pi$ in the environment.

We thus have two equivalent ways to describe the class of models which are PVE with respect to a set of policies $\Pi$.

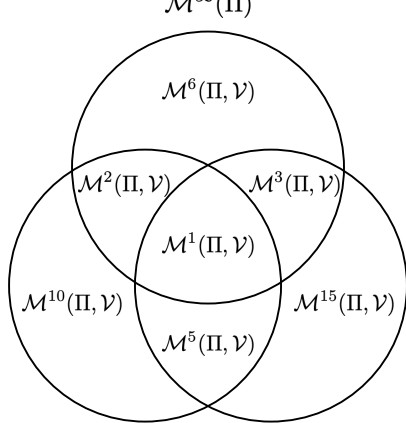

Figure 1: Topology of the space of order-$k$ VE classes. Given a set of policies $\Pi$, a set of functions $\mathcal{V}$ closed under Bellman updates, and $k, K \in \mathbb{Z}^+$ such that $k$ divides $K$, we have that $\mathcal{M}^k(\Pi, \mathcal{V}) \subseteq \mathcal{M}^K(\Pi, \mathcal{V})$.

The first, given in (5), is the order-$\infty$ limit of value equivalence with respect to $\Pi$ and the set of all functions $\mathbb{V}$. The second, given in (6), is the intersection of the classes of models that are order-$k$ VE with respect to the singleton policies in $\Pi$ and their respective value functions. This latter form is valid for *any* $k$, and will underpin our practical algorithmic instantiations of PVE.

Setting $k = 1$ in Proposition 2 we see that PVE can be written in terms of order-one VE. This means that $\mathcal{M}^\infty$ inherits many of the topological properties of $\mathcal{M}^1$ shown by Grimm et al. [17]. Specifically, we know that $\mathcal{M}'^\infty(\Pi) \subseteq \mathcal{M}^\infty(\Pi)$ if $\mathcal{M}' \subseteq \mathcal{M}$ and also that $\mathcal{M}^\infty(\Pi') \subseteq \mathcal{M}^\infty(\Pi)$ when $\Pi \subseteq \Pi'$ (these directly follow from Grimm et al.'s [17] Properties 1 and 3 respectively).

Proposition 2 also sheds further light into the relation between PVE and order-$k$ VE more generally. Let $\Pi$ be a set of policies and $\mathcal{V}_\pi$ their value functions. Then, for any $k \in \mathbb{Z}^+$, we have that

$$\mathcal{M}^k(\Pi, \mathcal{V}_\pi) = \bigcap_{\pi \in \Pi} \bigcap_{v \in \mathcal{V}_\pi} \mathcal{M}^k(\{\pi\}, \{v\}) \subseteq \bigcap_{\pi \in \Pi} \mathcal{M}^k(\{\pi\}, \{v_\pi\}) = \mathcal{M}^\infty(\Pi), \tag{7}$$

which is another way to say that $\mathcal{M}^\infty$ is, in some sense, the largest among all the VE classes. The reason why the size of VE classes is important is that it directly reflects the main motivation behind the VE principle. VE's premise is that models should be constructed taking into account their eventual use: if some aspects of the environment are irrelevant for value-based planning, it should not matter whether a model captures them or not. This means that all models that only differ with respect to these irrelevant aspects but are otherwise correct qualify as valid VE solutions. A larger VE class generally means that more irrelevant aspects of the environment are being ignored by the agent. We now make this intuition more concrete by showing how irrelevant aspects of the environment that are eventually captured by order-one VE are always ignored by PVE:

**Proposition 3.** *Let $\Pi \subseteq \mathbb{\Pi}$. If the environment state can be factored as $\mathcal{S} = \mathcal{X} \times \mathcal{Y}$ where $|\mathcal{Y}| > 1$ and $v_\pi(s) = v_\pi((x, y)) = v_\pi(x)$ for all $\pi \in \Pi$, then $\mathbb{M}^1(\Pi, \mathbb{V}) \subset \mathbb{M}^\infty(\Pi)$.*

Note that the subset relation appearing in Proposition 3 is strict. We can think of the variable '$y$' appearing in Proposition 3 as superfluous features that do not influence the RL task, like the background of an image or any other sensory data that is irrelevant to the agent's goal. A model is free to assign arbitrary dynamics to such irrelevant aspects of the state without affecting planning performance. Since order-one VE eventually pins down a model that describes everything about the environment, one would expect the size of $\mathcal{M}^\infty$ relative to $\mathcal{M}^1$ to increase as more superfluous features are added. Indeed, in our proof of Proposition 3 we construct a set of models in $\mathcal{M}^\infty(\mathbb{\Pi})$ which are in one-to-one correspondence with $\mathcal{Y}$, confirming this intuition (see Appendix A.2).

**Proper value equivalence yields models that are sufficient for optimal planning**

In general PVE does not collapse to a single model even in the limit of $\Pi = \mathbb{\Pi}$. At first this may cause the impression that one is left with the extra burden of selecting one among the PVE models. However, it can be shown that no such choice needs to be made:

**Proposition 4.** *An optimal policy for any $\tilde{m} \in \mathcal{M}^\infty(\mathbb{\Pi})$ is also an optimal policy in the environment.*

According to Proposition 1 any model $\tilde{m} \in \mathcal{M}^\infty(\mathbb{\Pi})$ used for planning will yield an optimal policy for the environment. In fact, in the spirit of ignoring as many aspects of the environment as possible, we can define an even larger PVE class by focusing on deterministic policies only:

**Corollary 1.** *Let $\mathbb{\Pi}_{det}$ be the set of all deterministic policies. An optimal policy for any $\tilde{m} \in \mathcal{M}^\infty(\mathbb{\Pi}_{det})$ is also optimal in the environment.*

Given that both $\mathcal{M}^\infty(\mathbb{\Pi})$ and $\mathcal{M}^\infty(\mathbb{\Pi}_{det})$ are sufficient for optimal planning, one may wonder if these classes are in fact the same. The following result states that the class of PVE models with respect to deterministic policies can be strictly larger than its counterpart defined with respect to all policies:

**Proposition 5.** *There exist environments and model classes for which $\mathcal{M}^\infty(\mathbb{\Pi}) \subset \mathcal{M}^\infty(\mathbb{\Pi}_{det})$.*

Figure 2 illustrates Proposition 5 with an example of environment and a model $\tilde{m}$ such that $\tilde{m} \in \mathcal{M}^\infty(\mathbb{\Pi}_{det})$ but $\tilde{m} \notin \mathcal{M}^\infty(\mathbb{\Pi})$.

To conclude our discussion on models that are sufficient for optimal planning, we argue that, in the absence of additional information about the environment or the agent, $\mathcal{M}^\infty(\mathbb{\Pi}_{det})$ is in fact the largest possible VE class that is guaranteed to yield optimal performance. To see why this is so, suppose we

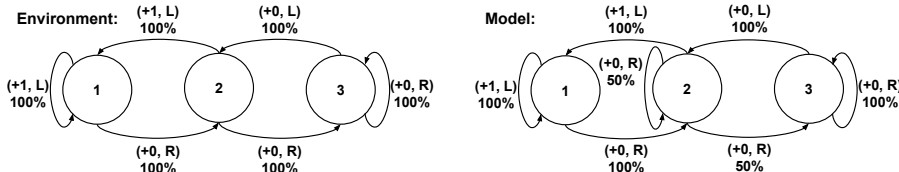

Figure 2: An environment / model pair with the same values for all deterministic policies but not all stochastic policies. The environment has three states and two actions: $\mathcal{A} = \{\mathrm{L}, \mathrm{R}\}$. The percentages in the figure indicate the probability of a given transition and the corresponding tuples $(r, a)$ indicate the reward associated with a given action. A deterministic policy cannot dither between $s_1$ and $s_3$ but a stochastic policy can. Note that the dynamics between the pair differs when taking action R from $s_2$. This difference will affect the dithering behavior of such a stochastic policy in a way that results in different model and environment values.

remove a single deterministic policy from $\mathbb{\Pi}_{\mathrm{det}}$ and pick an arbitrary model $\tilde{m} \in \mathcal{M}^\infty(\mathbb{\Pi}_{\mathrm{det}} - \{\pi\})$. Let $\tilde{v}_\pi$ be the value function of $\pi$ according to the model $\tilde{m}$. Because $\pi$ is not included in the set of policies used to enforce PVE, $\tilde{v}_\pi$ may not coincide with $v_\pi$, the actual value function of $\pi$ according to the environment. Now, if $\pi$ happens to be the only optimal policy in the environment and $\tilde{v}_\pi$ is not the optimal value function of $\tilde{m}$, the policy returned by this model will clearly be sub-optimal.

## 4 Learning a proper value-equivalent model

Having established that we want to find a model in $\mathcal{M}^\infty(\mathbb{\Pi}_{\mathrm{det}})$, we now turn our attention to how this can be done in practice. Following Grimm et al. [17], given a finite set of policies $\Pi$ and a finite set of functions $\mathcal{V}$, we cast the search for a model $\tilde{m} \in \mathcal{M}^k(\Pi, \mathcal{V})$ as the minimization of deviations from (4):

$$\ell_{\Pi,\mathcal{V}}^k(m^*, \tilde{m}) \equiv \sum_{\pi \in \Pi} \sum_{v \in \mathcal{V}} \|\mathcal{T}_\pi^k v - \tilde{\mathcal{T}}_\pi^k v\|, \tag{8}$$

where $\tilde{\mathcal{T}}_\pi$ are Bellman operators induced by $\tilde{m}$ and $\|\cdot\|$ is a norm.[2] Note that setting $k = \infty$ in (8) yields a loss that requires computing $\tilde{m}$'s value function—which is impractical to do if $\tilde{m}$ is being repeatedly updated. Thankfully, by leveraging the connection between order-$k$ VE and PVE given in Proposition 2, we can derive a practical PVE loss:

$$\ell_{\Pi,\infty}^k(m^*, \tilde{m}) \equiv \sum_{\pi \in \Pi} \|\mathcal{T}_\pi^k v_\pi - \tilde{\mathcal{T}}_\pi^k v_\pi\| = \sum_{\pi \in \Pi} \|v_\pi - \tilde{\mathcal{T}}_\pi^k v_\pi\|. \tag{9}$$

Interestingly, given a set of policies $\Pi$, minimizing (9) for any $k$ will result in a model $\tilde{m} \in \mathcal{M}^\infty(\Pi)$ (*cf.* Proposition 2). As we will discuss shortly, this property can be exploited to generate multiple loss functions that provide a richer learning signal in practical scenarios.

Contrasting loss functions (8) and (9) we observe an important fact: unlike with other order-$k$ VE classes, PVE requires actual value functions to be enforced in practice. Since value functions require data and compute to be obtained, it is reasonable to ask whether the benefits of PVE justify the associated additional burden. Concretely, one may ask whether the sample transitions and computational effort spent in computing the value functions to be used with PVE would not be better invested in enforcing other forms of VE over arbitrary functions that can be readily obtained.

We argue that in many cases one does not have to choose between order-$k$ VE and PVE. Value-based RL algorithms usually compute value functions iteratively, generating a sequence of functions $v_1, v_2, ...$ which will eventually converge to $\tilde{v}_\pi$ for some $\pi$. A model-based algorithm that computes $\tilde{v}_\pi$ in this way has to somehow interleave this process with the refinement of the model $\tilde{m}$. When it comes to VE, one extreme solution is to only use the final approximation $\tilde{v}_\pi \approx v_\pi$ in an attempt to enforce PVE through (9). It turns out that, as long as the sequence $v_1, v_2, ...$ is approaching $v_\pi$, one can use *all* the functions in the sequence to enforce PVE with respect to $\pi$. Our argument is based on the following result:

---

[2]We can also impose VE with infinite sets of functions and policies by replacing the respective sums with integrals; in this case one may consider taking a supremum over VE terms to avoid situations where VE is not necessarily satisfied on measure 0 sets.

**Proposition 6.** *For any $\pi \in \Pi$, $v \in \mathbb{V}$ and $k, n \in \mathbb{Z}^+$, we have that*

$$\|v_\pi - \tilde{\mathcal{T}}_\pi^k v_\pi\|_\infty \leq (\gamma^k + \gamma^n) \underbrace{\|v_\pi - v\|_\infty}_{\epsilon_v} + \underbrace{\|\mathcal{T}_\pi^n v - \tilde{\mathcal{T}}_\pi^k v\|_\infty}_{\epsilon_{ve}}. \tag{10}$$

Note that the left-hand side of (10) corresponds to one of the terms of the PVE loss (9) associated with a given $\pi$. This means that, instead of minimizing this quantity directly, one can minimize the upper-bound on the right-hand side of (10). The first term in this upper bound, $\epsilon_v$, is the conventional value-function approximation error that most value-based methods aim to minimize (either directly or indirectly). The second term, $\epsilon_{ve}$, is similar to the terms appearing in the order-$k$ VE loss (8), except that here the number of applications of $\mathcal{T}_\pi$ and of its approximation $\tilde{\mathcal{T}}_\pi$ do not have to coincide.

All the quantities appearing in $\epsilon_{ve}$ are readily available or can be easily approximated using sample transitions [17]. Thus, $\epsilon_{ve}$ can be used to refine the model $\tilde{m}$ using functions $v$ that are not necessarily value functions. As $v \to v_\pi$, two things happen. First, $\epsilon_{ve}$ approaches one of the terms of the PVE loss (9) associated with policy $\pi$. Second, $\epsilon_v$ vanishes. Interestingly, the importance of $\epsilon_v$ also decreases with $n$ and $k$, the number of times $\mathcal{T}_\pi$ and $\tilde{\mathcal{T}}_\pi$ are applied in $\epsilon_{ve}$, respectively. This makes sense: since $\mathcal{T}_\pi^n v \to v_\pi$ as $n \to \infty$ and, by definition, VE approaches PVE as $k \to \infty$, we have that $\epsilon_{ve}$ approaches the left-hand side of (10) as both $n$ and $k$ grow.

**An extended example: MuZero through the lens of value equivalence**

Grimm et al. [17] suggested that the VE principle might help to explain the empirical success of recent RL algorithms like Value Iteration Networks, the Predictron, Value Prediction Networks, TreeQN, and MuZero [38, 33, 24, 12, 31]. In this section we investigate this hypothesis further and describe a possible way to interpret one of these algorithms, MuZero, through the lens of VE. We acknowledge that the derivation that follows abstracts away many details of MuZero and involves a few approximations of its mechanics, but we believe it captures and explains the algorithm's essence.

MuZero is a model-based RL algorithm that achieved state-of-the-art performance across both board games, such as Chess and Go, and Atari 2600 games [31]. The model $\tilde{m}$ in MuZero is trained on sequences of states, actions and rewards resulting from executing a "behavior policy" in the environment: $s_{t:t+n+K}, a_{t:t+n+K}, r_{t:t+n+K}$ where $n$ and $K$ are hyperparameters of the agent which will be explained shortly. The agent produces an "agent state" $z_t^0$ from $s_t$ and subsequently generates $z_t^{1:K}$ by using its model to predict the next $K$ agent states following actions $a_{t:t+K}$. The agent also maintains reward and value function estimates as a function of agent states, which we denote $\tilde{r}(z)$ and $v(z)$ respectively. A variant[3] of MuZero's per-state model loss can thus be expressed as:

$$\ell^\mu(s_t) = \sum_{k=0}^{K} (V_{t+k} - v(z_t^k))^2 + (r_{t+k} - \tilde{r}(z_t^k)))^2 \tag{11}$$

where $V_{t+k} = r_{t+k} + \cdots + \gamma^{n-1} r_{t+k+n-1} + \gamma^n v^{targ}(z_{t+k+n}^0)$. The term $v^{targ}$ is a value target produced by Monte-Carlo tree search (MCTS, [7]). Because the behavior policy is itself computed via MCTS, we have that $v^{targ} \approx v$; for simplicity we will assume that $v^{targ} = v$ and only use $v$.

In what follows we show, subject to a modest smoothness assumption, that minimizing MuZero's loss with respect to its behavior policy, $\pi$, also minimizes a corresponding PVE loss. Put precisely:

$$C \cdot \mathbb{E}_{d_\pi}[\ell^\mu(S_t)] \geq \left(\ell_{\{\pi\},\infty}^K(m^*, \tilde{m})\right)^2 \tag{12}$$

for some $C > 0$, where $d_\pi$ is a stationary distribution. We proceed by combining two derivations: a lower-bound on $\mathbb{E}_{d_\pi}[\ell^\mu(S_t)]$ in (15), and an upper-bound on $(\ell_{\{\pi\},\infty}^K(m^*, \tilde{m}))^2$ in (17).

As a preliminary step we note that $\ell^\mu(s_t)$ and $\ell_{\{\pi\},\infty}^K(m^*, \tilde{m})$ are expressed in terms of samples and expectations respectively. We note the following connection between these quantities:

$$\mathbb{E}[r_{t+k}|s_t] = \mathcal{P}_\pi^k[r_\pi](s_t), \qquad \mathbb{E}[V_{t+k}|s_t] = \mathcal{P}_\pi^k \mathcal{T}_\pi^n[v_\pi](s_t),$$
$$\mathbb{E}[\tilde{r}(z_t^k)|s_t] = \tilde{\mathcal{P}}_\pi^k[\tilde{r}_\pi](s_t), \qquad \mathbb{E}[v(z_t^k)|s_t] = \tilde{\mathcal{P}}_\pi^k[v_\pi](s_t), \tag{13}$$

---

[3]In reality MuZero uses a categorical representation for its value and reward functions and minimizes them using a cross-entropy objective. We argue that this choice is not essential to its underlying ideas and use scalar representations with a squared loss to simplify our analysis.

where $\mathcal{P}_\pi^k$ is the $k$-step environment transition operator under policy $\pi$: $\mathcal{P}_\pi^k[x](s_t) = \mathbb{E}[x(S_{t+k})|s_t, m^*, \pi]$, $r_\pi(s) = \mathbb{E}_{A \sim \pi}[r(s, A)]$ and $\tilde{\mathcal{P}}_\pi^k$ and $\tilde{r}_\pi$ are the corresponding quantities using the model instead of the environment. The above expectations are taken with respect to the environment or model and $\pi$ as appropriate. We now derive our lower-bounds on $\mathbb{E}_{d_\pi}[\ell^\mu(S_t)]$:

$$
\mathbb{E}_{d_\pi}[\ell^\mu(S_t)] = \mathbb{E}_{d_\pi}\Big[ \sum_{k=0}^K \mathbb{E}[\,(V_{t+k} - v(z_t^k))^2 \mid S_t\,] + \sum_{k=0}^K \mathbb{E}[\,(r_{t+k} - \tilde{r}(z_t^k))^2 \mid S_t\,]\Big]
$$

$$
\geq \mathbb{E}_{d_\pi}\Big[ \sum_{k=0}^K (\mathbb{E}[\,V_{t+k} \mid S_t\,] - \mathbb{E}[\,v(z_t^k) \mid S_t\,])^2 + \sum_{k=0}^K (\mathbb{E}[\,r_{t+k} \mid S_t\,] - \mathbb{E}[\,\tilde{r}(z_t^k) \mid S_t\,])^2\Big]
$$

$$
= \sum_{k=0}^K \mathbb{E}_{d_\pi}\Big[ (\mathcal{P}_\pi^k \mathcal{T}_\pi^n v(S_t) - \tilde{\mathcal{P}}_\pi^k v(S_t))^2 \Big] + \sum_{k=0}^K \mathbb{E}_{d_\pi}\Big[ (\mathcal{P}_\pi^k r_\pi(S_t) - \tilde{\mathcal{P}}_\pi^k \tilde{r}_\pi(S_t))^2 \Big]
$$

(14)

where we apply the tower-property, Jensen's inequality and the identities in (13). We write the expression using norms and drop all terms except $k \in \{0, K\}$ in the first sum to obtain:

$$
\mathbb{E}_{d_\pi}[\ell^\mu(S_t)] \geq \|\mathcal{T}_\pi^n v - v\|_{d_\pi}^2 + \|\mathcal{P}_\pi^K \mathcal{T}_\pi^n v - \tilde{\mathcal{P}}_\pi^K v\|_{d_\pi}^2 + \sum_{k=0}^K \|\mathcal{P}_\pi^k r_\pi - \tilde{\mathcal{P}}_\pi^k \tilde{r}_\pi\|_{d_\pi}^2 \qquad (15)
$$

recalling that $\|x - y\|_{d_\pi}^2 = \mathbb{E}_{d_\pi}[(x(S_t) - y(S_t))^2]$. To derive an upper-bound for $(\ell_{\{\pi\},\infty}^K(m^*, \tilde{m}))^2$ we assume that the error in value estimation is smooth in the sense that there is some $g > 0$ (independent of $v$) such that $\|v - v_\pi\|_\infty < g \cdot \|v - v_\pi\|_{d_\pi}$. We can then use a modified version of (10) for the $d_\pi$-weighted $\ell_2$-norm (see Appendix A.2), plugging in $n + K$ and $K$:

$$
\|v_\pi - \tilde{\mathcal{T}}_\pi^K v_\pi\|_{d_\pi} \leq (g\gamma^K + \gamma^{n+K})\|v_\pi - v\|_{d_\pi} + \|\mathcal{T}_\pi^{K+n} v - \tilde{\mathcal{T}}_\pi^K v\|_{d_\pi}
$$

$$
\leq \gamma^K (g + \gamma^n)\|v_\pi - v\|_{d_\pi} + \|\mathcal{P}_\pi^K \mathcal{T}_\pi^n v - \tilde{\mathcal{P}}_\pi^K v\|_{d_\pi} + \sum_{k=0}^K \|\mathcal{P}_\pi^k r_\pi - \tilde{\mathcal{P}}_\pi^k \tilde{r}_\pi\|_{d_\pi}
$$

$$
\leq \gamma^K \frac{(g + \gamma^n)}{(1 - \gamma^n)}\|\mathcal{T}_\pi^n v - v\|_{d_\pi} + \|\mathcal{P}_\pi^K \mathcal{T}_\pi^n v - \tilde{\mathcal{P}}_\pi^K v\|_{d_\pi} + \sum_{k=0}^K \|\mathcal{P}_\pi^k r_\pi - \tilde{\mathcal{P}}_\pi^k \tilde{r}_\pi\|_{d_\pi},
$$

(16)

from here we can square both sides and apply Jensen's inequality,

$$
\|v_\pi - \tilde{\mathcal{T}}_\pi^K v_\pi\|_{d_\pi}^2 \leq ab\|\mathcal{T}_n^\pi v - v\|_{d_\pi}^2 + b\|\mathcal{P}_\pi^K \mathcal{T}_\pi^n v - \tilde{\mathcal{P}}_\pi^K v\|_{d_\pi}^2 + b\sum_{k=0}^K \|\mathcal{P}_\pi^k r_\pi - \tilde{\mathcal{P}}_\pi^k \tilde{r}_\pi\|_{d_\pi}^2, \quad (17)
$$

where $a = \gamma^K (g + \gamma^n)(1 - \gamma^n)^{-1}$ and $b = a + K + 2$. Combining (17) and (15) we obtain:

$$
ab \cdot \mathbb{E}_{d_\pi}[\ell^\mu(S_t)] \geq \|v_\pi - \tilde{\mathcal{T}}_\pi^K v_\pi\|_{d_\pi}^2 = \left(\ell_{\{\pi\},\infty}^K(m^*, \tilde{m})\right)^2, \qquad (18)
$$

thus minimizing MuZero's loss minimizes a squared PVE loss with respect to a single policy.

## 5   Experiments

We first provide results from tabular experiments on a stochastic version of the Four Rooms domain which serve to corroborate our theoretical claims. Then, we present results from experiments across the full Atari 57 benchmark [3] showcasing that the insights from studying PVE and its relationship to MuZero can provide a benefit in practice at scale. See Appendix A.3 for a full account of our experimental procedure.

In Section 3 we described the topological relationships between order-$k$ and PVE classes. This is summarized by Proposition 1, which shows that, for appropriately defined $\mathcal{V}$ and $\Pi$, $\mathcal{M}^k \subseteq \mathcal{M}^K$ if $K$ is a multiple of $k$. We illustrate this property empirically by randomly initializing a set of models and then using (8) (or (9) for the limiting case of $k = \infty$) to iteratively update them towards $\mathcal{M}^k(\Pi, \mathcal{V})$, with $k \in \{1, 30, 40, 50, 60, \infty\}$. We take the vectors representing these models and project them onto their first two principal components in order to visualise their paths through learning. The results are

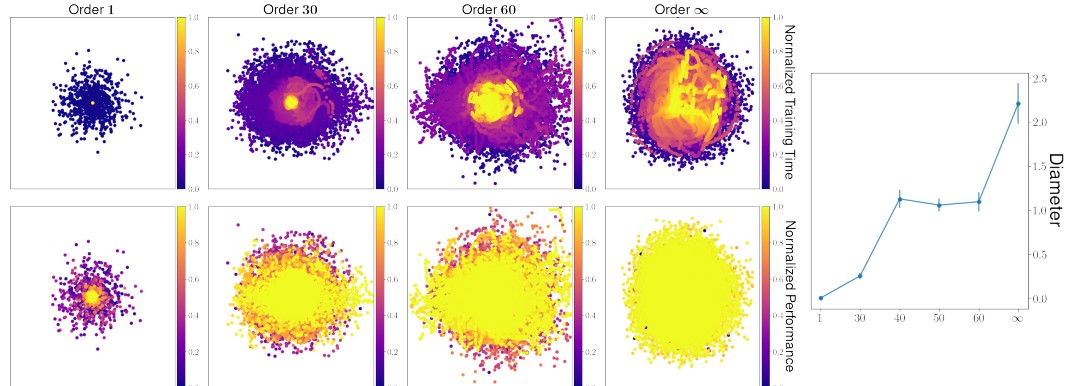

Figure 3: All scatter plots are generated by tracking the training progress over 500,000 iterations of models with different order-$k$ VE objectives. In each plot 120 models were tracked; at every 1000 timesteps the full set of models is converted into vector form and projected onto their first two principal components before being plotted (details in the appendix). **Top row:** points are colored according to their progress through training. **Bottom row:** points are colored according to the average value of the associated model's optimal policy on the environment. **Rightmost plot:** line-plot of the diameters of these scatter plots against the model-class order.

shown on the top row of Figure 3. In accordance with the theory, we see that the space of converged models, represented with the brightest yellow regions, grows with $k$. This trend is summarised in the rightmost plot, which shows the diameter of the scatter plots for each $k$. In the bottom row of Figure 3 we use color to show the value that the optimal policy of each model achieves in the true environment. As predicted by our theory, the space of models that are sufficient for optimal planning also grows with $k$.

Model classes containing many models with optimal planning performance are particularly advantageous when the set of models that an agent can represent is restricted, since the larger the set of suitable models the greater the chance of an overlap between this set and the set of models representable by the agent. Proposition 4 and Corollary 1 compared $\mathcal{M}^\infty(\mathbb{\Pi})$ and $\mathcal{M}^\infty(\mathbb{\Pi}_{\text{det}})$, showing that, although any model in either class is sufficient for planning, $\mathcal{M}^\infty(\mathbb{\Pi}) \subseteq \mathcal{M}^\infty(\mathbb{\Pi}_{\text{det}})$. This suggests that it might be better to learn a model in $\mathcal{M}^\infty(\mathbb{\Pi}_{\text{det}})$ when the agent has limited capacity.

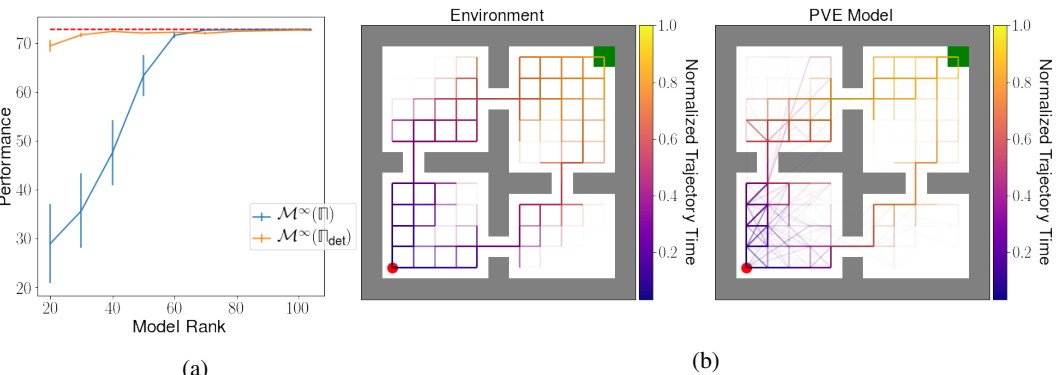

Figure 4: **(a)** Comparison of the performance of optimal policies obtained from capacity constrained models trained to be in $\mathcal{M}^\infty(\mathbb{\Pi}_{\text{det}})$ and $\mathcal{M}^\infty(\mathbb{\Pi})$. For each action $a \in \mathcal{A}$, the transition dynamics $\tilde{P}_a$ is constrained to have a rank of at most $k$. The red dashed line represents the performance of the optimal environment policy. **(b)** Trajectories starting from the bottom-right state (red dot) sampled from the optimal environment policy in both the environment and a PVE model. Note the numerous diagonal transitions in the PVE model which are not permitted in the environment.

We illustrate that this is indeed the case in Figure 4b. We progressively restrict the space of models that the agent can represent and attempt to learn models in either $\mathcal{M}^\infty(\mathbb{\Pi})$ or $\mathcal{M}^\infty(\mathbb{\Pi}_{\text{det}})$. Indeed, we find that the larger class, $\mathcal{M}^\infty(\mathbb{\Pi}_{\text{det}})$, yields superior planning performance as agent capacity decreases.

Given their importance, we provide intuition on the ways that *individual* PVE models differ from the environment. In Figure 4a we compare trajectories starting at the same initial state (denoted by a red-circle) from the optimal environment policy in both the environment and in a randomly sampled model from $\mathcal{M}^\infty(\mathbb{\Pi})$. In the PVE model there are numerous diagonal transitions not permitted by the environment. Note that while the PVE model has very different dynamics than the environment, these differences must "balance out", as it still has the same values under any policy as the environment.

In Section 4 we showed that minimizing Muzero's loss function is analogous to minimizing an upper-bound on a PVE loss (9) with respect to the agent's current policy $\pi$—which corresponds to finding a model in $\mathcal{M}^\infty(\Pi)$ where $\Pi = \{\pi\}$. Note that our guarantee on the performance of PVE models (Corollary 1) holds when $\Pi$ contains all deterministic policies. While it is not feasible to enforce $\Pi = \mathbb{\Pi}_{\text{det}}$, we can use previously seen policies by augmenting the MuZero algorithm with a buffer of past policies and their approximate value functions (we do so by periodically storing the corresponding parameters). We can then add an additional loss to MuZero with the form of the original value loss but using the past value functions. We still use sampled rewards to construct value targets, but use the stored policies to compute off-policy corrections using V-trace [9].

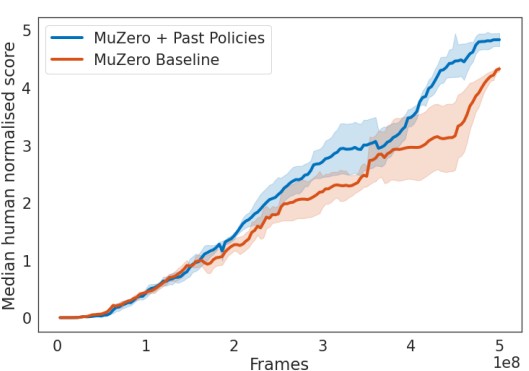

Figure 5: Comparison of our proposed modification to MuZero with an unmodified baseline.

To test this proposal we use an on policy (i.e., without a replay buffer) implementation of MuZero run for 500M frames (as opposed to 20B frames in the online result of [31]) on the Atari 57 benchmark and find that using our additional loss yields an advantage in the human normalised median performance shown in Figure 5. MuZero's data efficiency can also be improved with the aid of a replay buffer of trajectories from past policies [32], which may also capture some of the advantages of expanding the set of policies used for PVE.

## 6   Related work

Our work is closely related to the value-aware model learning (VAML, IterVAML, [11, 10]) which learns models to minimize the discrepancy between their own Bellman optimality operators and the environment's on the optimal value function—an a priori unknown quantity. To handle this VAML specifies a family of potential value functions and minimizes the worst-case discrepancy across them, whereas IterVAML minimizes the discrepancy with respect to model's current estimate of the value function in a value iteration inspired scheme. PVE and the VAML family are complementary works, with VAML addressing its induced optimization problems and PVE addressing its induced model classes. Both, however, advocate for learning models with their eventual use in mind—a view that is aligned with many criticisms of the maximum likelihood objective for model learning [11, 21, 2, 23]).

It is worth mentioning the relationship between PVE and TD models [35] which, for a given policy, defines any $R \in \mathbb{R}^{|\mathcal{S}|}$ and $P \in \mathbb{R}^{|\mathcal{S}| \times |\mathcal{S}|}$ with $\lim_{k \to \infty} P^k = 0$ as a *valid model* if $V = R + P^\top V$ where $V \in \mathbb{R}^{|\mathcal{S}|}$ represents $v_\pi$. Clearly all models in $\mathcal{M}^\infty(\{\pi\})$ are valid models, however, since $P$ is not necessarily a transition matrix, the converse does not hold. While TD models are restricted to prediction rather than control, their generality warrants further inquiry.

Order-$k$ and PVE model classes form equivalences between MDPs and thus can be situated among other equivalence notions which can be formulated as state-aggregations [8, 26, 25, 16, 28, 13, 34, 39, 5, 40]. As pointed out by Grimm et al. [17], the interaction between arbitrary state-aggregation

and models can be captured with special cases of order-one VE. Our extension of higher-order VEs potentially offers the possibility of "blending" existing notions of state aggregation with PVE.

A notable instance of state-aggregation is bisimulation [22], which uses a relation to aggregate states that have the same immediate rewards and transition dynamics into other aggregated states. Bisimulation metrics [13] provide smooth measures of how closely pairs of states satisfy bisimulation relations. These concepts have become increasingly popular in deep reinforcement learning where they are used to guide the learning of effective representations [43, 42, 15, 1]. Although both bisimulation and PVE provide direction for learning internal aspects of an agent, they are fundamentally different in their purview—bisimulation concerns representation learning, while PVE concerns the learning of models given a representation of state.

Beyond bisimulation, representation learning has a wide literature [41, 20, 6, 14, 4] including several modern works which explicitly study the conjunction of model learning with state representation [44, 42, 15]. These are further complemented by efforts to learn state representations and models jointly in the service of value-based planning [12, 33, 24, 18, 31, 38].

## 7    Conclusion and future work

We extended the value equivalence principle by defining a spectrum of order-$k$ VE sets in which models induce the same $k$-step Bellman operators as the environment. We then explored the topology of the resulting equivalence classes and defined the limiting class when $k \to \infty$ as PVE. If a model is PVE to the environment with respect to a set of policies $\Pi$, then the value functions of all policies in $\Pi$ are the same in the environment and the model. The fact that PVE classes can be defined using only a set of policies eliminates the need for specifying a set of functions to induce VE—resolving a fundamental issue left open by Grimm et al. [17]. Importantly, we showed that being PVE with respect to all deterministic policies is sufficient for a model to plan optimally in the environment. In the absence of additional information, this is the largest possible VE class that yields optimal planning. On the practical side, we showed how the MuZero algorithm can be understood as minimizing an upper bound on a PVE loss, and leveraged this insight to improve the algorithm's performance.

Though our efforts have advanced the understanding of value equivalence and proven useful algorithmically, there is still work to be done in developing a VE theory whose assumptions hold in practice. This remaining work can be broadly grouped into two areas (1) understanding the role of approximation in VE and (2) establishing performance guarantees for VE models with arbitrary sets of policies and functions. We leave these as future work.

**Acknowledgements**

We thank Angelos Filos and Sonya Kotov for many thought-provoking discussions. Christopher Grimm's work was made possible by the support of the Lifelong Learning Machines (L2M) grant from the Defense Advanced Research Projects Agency. Any opinions, findings, conclusions, or recommendations expressed here are those of the authors and do not necessarily reflect the views of the sponsors.

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
