# A Appendix

## A.1 Illustrative MDPs

Several proofs in A.2 rely on constructing special MDPs to serve as examples or counterexamples. We reserve this section to describe these MDPs for later reference.

### A.1.1 Ring and false-ring MDPs

We consider a simple $n$-state, 1 action "ring" MDP (Figure 6) denoted $m_\circ^n = (r, p)$ where:

$$r(s_i) = g(i) \ \forall i \in [n], \qquad p(s_{i+1}|s_i) = 1 \ \forall i \in [n-1] \ \text{ and } \ p(s_1|s_n) = 1 \tag{19}$$

where $g : i \mapsto \mathbb{R}$ is some function that defines the reward from transitioning away from state $i$. Since $|\mathcal{A}| = 1$ we omit actions from the reward and transition dynamics.

For each ring MDP and function $g$ we additionally construct a corresponding "false-ring" MDP (Figure 6) with the same state and actions spaces as $m_\circ^n$ but with states that only self-transition and with rewards designed to mimic the discounted $n$-step returns on Ring MDPs. We represent these as $\tilde{m}_\circ^n = (\tilde{r}, \tilde{p})$ where

$$\tilde{r}(s_i) = \frac{r^n(s_i)}{\sum_{t=0}^{n-1} \gamma^t}, \qquad \tilde{p}(s_i|s_i) = 1 \tag{20}$$

and $r^n(s_i)$ denotes the discounted $n$-step return starting from $s_i$ in $m_\circ^n$. Note that the discounted $n$-step return of an $n$-state false-ring MDP is the same as that of an $n$-state ring MDP.

We now provide some basic results about pairs of ring and false-ring MDPs that we will use periodically in our proofs.

**Lemma 1.** *For any $n \in \mathbb{Z}^+ \cup \{\infty\}$ if we treat the ring MDP $m_\circ^n$ as the environment and assume $\tilde{m}_\circ^n \in \mathcal{M}$ it follows that*

$$\tilde{m}_\circ^n \in \mathcal{M}^n(\mathbb{\Pi}, \mathbb{V}). \tag{21}$$

*when $n < \infty$ and*

$$\tilde{m}_\circ^n \in \mathcal{M}^\infty(\mathbb{\Pi}) \tag{22}$$

*when $n = \infty$.*

*Proof.* First we note that, since ring and false-ring MDPs only have one action, we can write $\mathbb{\Pi} = \{\pi\}$ where $\pi$ takes this action at all states. We first consider the case when $n < \infty$, noting that that both MDPs are deterministic and that for any state $s$, performing $n$ transitions will always return to $s$. We now consider an application of $n$-step Bellman operator of the false-ring model to an arbitrary function $v \in \mathbb{V}$:

$$
\begin{aligned}
\tilde{\mathcal{T}}_\pi^n v(s) &= \tilde{r}(s)\left(\sum_{t=0}^{n-1} \gamma^t\right) + \gamma^n v(s) \\
&= r^n(s)\left(\sum_{t=0}^{n-1} \gamma^t\right)^{-1}\left(\sum_{t=0}^{n-1} \gamma^t\right) + \gamma^n v(s) \\
&= r^n(s) + \gamma^n v(s) \\
&= \mathcal{T}_\pi^n v(s)
\end{aligned}
\tag{23}
$$

implying that $\tilde{m}_\circ^n \in \mathcal{M}^n(\mathbb{\Pi}, \mathbb{V})$ as needed. We now consider the case when $n = \infty$: here, we note that for any state $s \in \mathcal{S}$:

$$\tilde{r}(s) = \frac{r^\infty(s)}{\sum_{t=0}^\infty \gamma^t} = (1 - \gamma)v_\pi(s). \tag{24}$$

we can then write:

$$\tilde{v}_\pi(s) = \sum_{t=0}^\infty \gamma^t \tilde{r}(s) = (1 - \gamma)^{-1}(1 - \gamma)v_\pi(s) = v_\pi(s) \tag{25}$$

since $\tilde{m}_\circ^\infty$ only self-transitions at each state. This shows that $\tilde{m}_\circ^\infty \in \mathcal{M}^\infty(\Pi)$ as needed.

$\square$

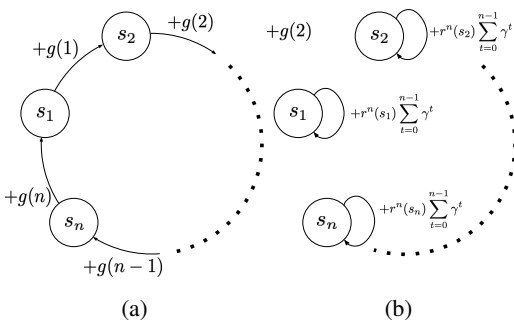

Figure 6: Ring and false-ring environments with reward structure defined by $g : \mathbb{Z}^+ \mapsto \mathbb{R}$. States are numbered circles and outgoing arrows indicate possible transitions from each state. Arrows are labeled by the reward attained from performing their transition.

**Lemma 2.** *Fix any* $k, K \in \mathbb{Z}^+ \cup \{\infty\}$ *with* $k < K$ *and let* $f : \mathcal{S} \mapsto \mathbb{R}$ *be any constant function. Let* $m = m_\circ^K$ *and* $\tilde{m} = \tilde{m}_\circ^K$. *For any* $\gamma \in (0, 1)$ *it follows that*

$$\mathcal{T}_\pi^k f(s_1) \neq \tilde{\mathcal{T}}_\pi^k f(s_1) \tag{26}$$

*where* $m_\circ^K$ *and* $\tilde{m}_\circ^K$ *are* $K$*-state ring and false-ring MDPs with* $g(i) = \mathbf{1}\{i \in [1, k]\}$.

*Proof.* We begin by examining the $k$-step Bellman operator and Bellman fixed-point under the ring $m_\circ^K$:

$$\mathcal{T}_\pi^k f(s_1) = r^k(s_1) + \gamma^k f(s_1) = r^K(s_1) + \gamma^k f(s_1) \tag{27}$$

where the second equality follows from the fact that $g$ ensures that no reward is received after the first $k$ steps from $s_1$.

Next we examine the corresponding $k$-step Bellman operator under the false-ring $\tilde{m}_\circ^K$:

$$\tilde{\mathcal{T}}_\pi^k f(s_1) = \tilde{r}^k(s_1) + \gamma^k f(s_1) = r^K(s_1)(\sum_{t=0}^{K-1} \gamma^t)^{-1} \sum_{t=0}^{k-1} \gamma^t + \gamma^k f(s_1) \tag{28}$$

where the second equality follows from the construction of $K$-step false-ring MDPs to match the $K$-step returns of their corresponding ring MDP.

Taken together Eqs. (27-28) imply that in order for $\mathcal{T}_\pi^k f(s_1) = \tilde{\mathcal{T}}_\pi^k f(s_1)$ it must be the case that $\sum_{t=0}^{K-1} \gamma^t = \sum_{t=0}^{k-1} \gamma^t$ which can only happen when $\gamma = 0$. Note that these properties hold when $K = \infty$. This completes the proof. $\square$

## A.2 Proofs

In this section we provide proofs of the results in the main text.

**Proposition 1.** *Let* $\mathcal{V}$ *be a set of functions such that if* $v \in \mathcal{V}$ *then* $\mathcal{T}_\pi v \in \mathcal{V}$ *for all* $\pi \in \Pi$. *Then, for* $k, K \in \mathbb{Z}^+$ *such that* $k$ *divides* $K$, *it follows that:*

*(i) For any* $\mathcal{M} \subseteq \mathbb{M}$ *and any* $\Pi \subseteq \mathbb{\Pi}$, *we have that* $\mathcal{M}^k(\Pi, \mathcal{V}) \subseteq \mathcal{M}^K(\Pi, \mathcal{V})$.

*(ii) If* $\Pi$ *is non-empty and* $\mathcal{V}$ *contains at least one constant function, then there exist environments such that* $\mathbb{M}^k(\Pi, \mathcal{V}) \subset \mathbb{M}^K(\Pi, \mathcal{V})$.

*Proof.* Consider some $m \in \mathcal{M}^k(\Pi, \mathcal{V})$. For any $\pi \in \Pi$ and $v \in \mathcal{V}$ we know that $\tilde{\mathcal{T}}_\pi^k v = \mathcal{T}_\pi^k v$. Since $k$ divides $K$ we know that $K = zk$ where $z \in \mathbb{Z}^+$. Hence

$$\mathcal{T}_\pi^K v = \underbrace{\mathcal{T}_\pi \cdots \mathcal{T}_\pi}_{K \text{ times}} v = \underbrace{\mathcal{T}_\pi^k \cdots \mathcal{T}_\pi^k}_{z \text{ times}} v \tag{29}$$

Finally since $\mathcal{V}$ is closed under Bellman updates we can write $\tilde{\mathcal{T}}_\pi^k v = \mathcal{T}_\pi^k v \in \mathcal{V}$, which allows us iteratively equate $k$-step environment and model operators on the right-hand side of Eq. (29) to obtain:

$$\underbrace{\mathcal{T}_\pi^k \cdots \mathcal{T}_\pi^k}_{z \text{ times}} v = \underbrace{\tilde{\mathcal{T}}_\pi^k \cdots \tilde{\mathcal{T}}_\pi^k}_{z \text{ times}} v = \tilde{\mathcal{T}}_\pi^K v. \tag{30}$$

This suffices to show that $m \in \mathcal{M}^K(\Pi, \mathcal{V})$ which means $\mathcal{M}^k(\Pi, \mathcal{V}) \subseteq \mathcal{M}^K(\Pi, \mathcal{V})$.

We now assume that $\mathcal{V}$ contains at least one constant function and $\Pi$ is non-empty and produce an instance of an environment and model class where the relation is strict. Let the environment be a $K$-state ring environment (see A.1.1): $m_\circ^K$ with $g(i) = \mathbf{1}\{i \in [1, k]\}$ and let $\mathcal{M} = \mathbb{M}$. Next we introduce a model given by the corresponding false-ring MDP (see A.1.1) $\tilde{m}_\circ^K$. From Lemma 1 we have that $\tilde{m}_\circ^K \in \mathcal{M}^K(\Pi, \mathcal{V})$.

Since there is at least one constant function $f \in \mathcal{V}$ we know that $\mathcal{T}_\pi^k f(s_1) \neq \tilde{\mathcal{T}}_\pi^k f(s_1)$ from Lemma 2. This is sufficient to show that $\tilde{m}_\circ^K \notin \mathcal{M}^k(\Pi, \mathcal{V})$ and thus we have proven that there are instances where $\mathcal{M}^k(\Pi, \mathcal{V}) \subset \mathcal{M}^K(\Pi, \mathcal{V})$. $\qquad\square$

**Proposition 2.** *For any $\Pi \subseteq \mathbb{\Pi}$ and any $k \in \mathbb{Z}^+$ it follows that*

$$\mathcal{M}^\infty(\Pi) = \bigcap_{\pi \in \Pi} \mathcal{M}^k(\{\pi\}, \{v_\pi\}), \tag{6}$$

*Proof.* We first note $\mathcal{M}^\infty(\Pi) = \bigcap_{\pi \in \Pi} \mathcal{M}^\infty(\{\pi\})$ and consider any $m \in \mathcal{M}^\infty(\{\pi\})$ for some $\pi \in \Pi$. From the definition of PVE we know $\tilde{v}_\pi = v_\pi$ and thus can say:

$$\begin{aligned}
& \tilde{v}_\pi = v_\pi \\
\implies & \tilde{\mathcal{T}}_\pi^k \tilde{v}_\pi = \tilde{\mathcal{T}}_\pi^k v_\pi \\
\implies & \tilde{v}_\pi = \tilde{\mathcal{T}}_\pi^k v_\pi \\
\implies & v_\pi = \tilde{\mathcal{T}}_\pi^k v_\pi \\
\implies & \mathcal{T}_\pi^k v_\pi = \tilde{\mathcal{T}}_\pi^k v_\pi
\end{aligned} \tag{31}$$

which suggests that $m \in \mathcal{M}^k(\{\pi\}, \{v^\pi\})$ and thus $\mathcal{M}^\infty(\{\pi\}) \subseteq \mathcal{M}^k(\{\pi\}, \{v^\pi\})$.

We now consider any element $m \in \mathcal{M}^k(\{\pi\}, \{v^\pi\})$, and note that from the definition of order-$k$ VE we know that $\tilde{\mathcal{T}}_\pi^k v_\pi = \mathcal{T}_\pi^k v_\pi$, thus we can say:

$$\begin{aligned}
& \tilde{\mathcal{T}}_\pi^k v_\pi = \mathcal{T}_\pi^k v_\pi \\
\implies & \tilde{\mathcal{T}}_\pi^k v_\pi = v_\pi \\
\implies & \tilde{\mathcal{T}}_\pi^{2k} v_\pi = \tilde{\mathcal{T}}_\pi^k v_\pi \\
\implies & \tilde{\mathcal{T}}_\pi^{2k} v_\pi = v_\pi
\end{aligned} \tag{32}$$

where we can repeat the process described in these implications ad-infinitum to obtain $\tilde{v}_\pi = \lim_{n \to \infty} \tilde{\mathcal{T}}_\pi^{nk} v_\pi = v_\pi$. Hence $m \in \mathcal{M}^\infty(\{\pi\})$ and thus $\mathcal{M}^k(\{\pi\}, \{v^\pi\})$.

Taken together this shows that $\mathcal{M}^\infty(\{\pi\}) = \mathcal{M}^k(\{\pi\}, \{v^\pi\})$ for any $k$ and $\pi$ thus:

$$\mathcal{M}^\infty(\Pi) = \bigcap_{\pi \in \Pi} \mathcal{M}^\infty(\{\pi\}) = \bigcap_{\pi \in \Pi} \mathcal{M}^k(\{\pi\}, \{v^\pi\}) \tag{33}$$

for any $k \in \mathbb{Z}^+$. $\qquad\square$

**Corollary 2.** *Let $\Pi \subseteq \mathbb{\Pi}$ and let $\mathcal{V}$ be as in Proposition 1 for $k \in \mathbb{Z}^+$ then we have that $\mathcal{M}^k(\Pi, \mathcal{V}) \subseteq \mathcal{M}^\infty(\Pi)$. Moreover, if $\Pi$ is non-empty and $\mathcal{V}$ contains at least one constant function, then there exist environments such that $\mathbb{M}^k(\Pi, \mathcal{V}) \subset \mathbb{M}^\infty(\Pi)$*

*Proof.* Consider some $m \in \mathcal{M}^k(\Pi, \mathcal{V})$. From the generalization of Property 1 we know that $m \in \mathcal{M}^{zk}(\Pi, \mathcal{V})$ for any $z \in \mathbb{Z}^+$ since $k$ divides $zk$. Thus we know that $\tilde{\mathcal{T}}_\pi^{zk} v = \mathcal{T}_\pi^{zk} v$ for any

choice of $\pi \in \Pi$, $v \in \mathcal{V}$ and $z \in \mathbb{Z}^+$. Accordingly the expressions are equal in the limit as $z \to \infty$. Combining this with the fact that both $\tilde{\mathcal{T}}_\pi$ and $\mathcal{T}_\pi$ are contraction mappings, we obtain:

$$\tilde{v}_\pi = \lim_{z \to \infty} \tilde{\mathcal{T}}_\pi^{zk} v = \lim_{z \to \infty} \mathcal{T}_\pi^{zk} v = v_\pi \tag{34}$$

which implies $m \in \mathcal{M}^\infty(\Pi)$ and thus $\mathcal{M}^k(\Pi, \mathcal{V}) \subseteq \mathcal{M}^\infty(\Pi)$, as needed.

Moreover, so long that $\Pi$ is nonempty and $\mathcal{V}$ contains some constant function $f$, we can construct a pair of $\infty$-state ring / false-ring MDPs: $m_\circ^\infty$ and $\tilde{m}_\circ^\infty$ with $g(i) = \mathbf{1}\{i \in [1, k]\}$ (see A.1.1). By assuming that $m_\circ^\infty$ is the environment, Lemma 1 tells us that $\tilde{m}_\circ^\infty \in \mathcal{M}^\infty(\Pi)$ and we know from Lemma 2 that $\mathcal{T}_\pi^k f(s_1) \neq \tilde{\mathcal{T}}_\pi^k f(s_1)$ hence $\tilde{m}_\circ^\infty \notin \mathcal{M}^k(\Pi)$. $\qquad\square$

**Proposition 3.** *Let $\Pi \subseteq \mathbb{\Pi}$. If the environment state can be factored as $\mathcal{S} = \mathcal{X} \times \mathcal{Y}$ where $|\mathcal{Y}| > 1$ and $v_\pi(s) = v_\pi((x, y)) = v_\pi(x)$ for all $\pi \in \Pi$, then $\mathbb{M}^1(\Pi, \mathbb{V}) \subset \mathbb{M}^\infty(\Pi)$.*

*Proof.* Assume that $\mathcal{M} = \mathbb{M}$. Denote the environment reward and transition dynamics as $(r, p)$. For any value $y_0 \in \mathcal{Y}$ we consider a model $m_{y_0}$:

$$\begin{aligned} r_{m_{y_0}}((x, y), a) &= r((x, y), a) \\ p_{m_{y_0}}((x', y')|(x, y), a) &= \mathbf{1}\{y' = y_0\}p(x'|(x, y), a). \end{aligned} \tag{35}$$

We now examine the Bellman fixed-point induced by environment for any policy $\pi \in \Pi$:

$$\begin{aligned} v_\pi((x, y)) &= \int_\mathcal{A} \pi(a|(x, y))r((x, y), a) + \gamma \int_\mathcal{X} \int_\mathcal{Y} p((x', y')|(x, y), a)v_\pi((x', y'))dx'dy'da \\ &= \int_\mathcal{A} \pi(a|(x, y))r((x, y), a) + \gamma \int_\mathcal{X} \int_\mathcal{Y} p(x'|(x, y), a)p(y'|x', (x, y), a)v_\pi(x')dx'dy'da \\ &= \int_\mathcal{A} \pi(a|(x, y))r((x, y), a) + \gamma \int_\mathcal{X} p(x'|(x, y), a)v_\pi(x')dx'da. \end{aligned} \tag{36}$$

We can compare this to the Bellman operator induced by our model for the same policy:

$$\begin{aligned} \tilde{\mathcal{T}}_\pi v((x, y)) &= \int_\mathcal{A} \pi(a|(x, y))r((x, y), a) + \gamma \int_\mathcal{X} \int_\mathcal{Y} \mathbf{1}\{y' = y_0\}p(x'|(x, y), a)v((x, y))dx'dy'da \\ &= \int_\mathcal{A} \pi(a|(x, y))r((x, y), a) + \gamma \int_\mathcal{X} p(x'|(x, y), a)v((x', y_0))dxda \end{aligned} \tag{37}$$

Notice that $v_\pi$ is a fixed point of this operator, hence $\tilde{v}_\pi = v_\pi$ and and thus $m_{y_0} \in \mathcal{M}^\infty(\Pi)$ (since our particular choice of $\pi \in \Pi$ was arbitrary). Moreover, we can construct different models for each $y_0 \in \mathcal{Y}$, we know that

$$\mathcal{M}_\mathcal{Y} = \{m_y : y \in \mathcal{Y}\} \subseteq \mathcal{M}^\infty(\mathbb{\Pi}). \tag{38}$$

Moreover, suppose $m_{y_0} \in \mathcal{M}^1(\Pi, \mathbb{V})$. This implies that for all $v \in \mathbb{V}$

$$\begin{aligned} &\tilde{\mathcal{T}}_\pi v((x, y)) = \mathcal{T}_\pi v((x, y)) \\ \implies &\int_\mathcal{A} \int_\mathcal{X} \int_\mathcal{Y} \pi(a|(x, y))p_{m_{y_0}}((x', y')|(x, y), a)v((x', y'))dx'dy'da \\ &= \int_\mathcal{A} \int_\mathcal{X} \int_\mathcal{Y} \pi(a|(x, y))p((x', y')|(x, y), a)v((x', y'))dx'dy'da \\ \implies &\int_\mathcal{A} \int_\mathcal{X} \pi(a|(x, y))p(x'|(x, y), a)v((x', y_0))dx' \\ &= \int_\mathcal{A} \int_\mathcal{X} \int_\mathcal{Y} \pi(a|(x, y))p((x', y')|(x, y), a)v((x', y'))dx'dy' \end{aligned} \tag{39}$$

we now choose $v((x, y)) = \mathbf{1}\{y \neq y_0\}$ which reduces the above equations to:

$$\begin{aligned} \implies 0 &= \int_\mathcal{A} \int_\mathcal{X} \int_{\mathcal{Y} \neq y_0} \pi(a|(x, y))p((x', y')|(x, y), a) = \mathbb{P}(y' \neq y_0|x, y, \pi) \\ \implies &\mathbb{P}(y' = y_0|x, y, \pi) = 1 \end{aligned} \tag{40}$$

where $\mathbb{P}$ denotes the conditional probability of an event.

Now consider the class of models defined by Eq. (38). Suppose $\mathcal{M}_{\mathcal{Y}} \in \mathcal{M}^1(\Pi, \mathbb{V})$, by Eq. (40) this would mean that $\mathbb{P}(y' = y_0 | x, y, \pi) = 1$ for all $y_0 \in \mathcal{Y}$. This is impossible unless $|\mathcal{Y}| = 1$ hence there must exist $m_{y_0} \notin \mathcal{M}^1(\Pi, \mathbb{V})$ and thus $\mathcal{M}^1(\Pi, \mathbb{V}) \subset \mathcal{M}^\infty(\Pi, \mathbb{V})$.

$\square$

**Corollary 1.** *Let $\Pi_{det}$ be the set of all deterministic policies. An optimal policy for any $\tilde{m} \in \mathcal{M}^\infty(\Pi_{det})$ is also optimal in the environment.*

*Proof.* Denote a deterministic optimal policy with respect to the environment as $\pi^*$. Let $\tilde{m} \in \mathcal{M}^\infty(\Pi_{\det})$ and $\tilde{\pi}^*$ be a deterministic optimal policy with respect $\tilde{m}$.

Suppose $\tilde{\pi}^*$ were not optimal in the environment. This implies that $v_{\pi^*}(s) \geq v_{\tilde{\pi}^*}(s) \forall s \in \mathcal{S}$ with strict inequality for at least one state. However, since $\pi^*$ and $\tilde{\pi}^*$ are deterministic we have:

$$\tilde{v}_{\pi^*}(s) = v_{\pi^*}(s) > v_{\tilde{\pi}^*}(s) = \tilde{v}_{\tilde{\pi}^*}(s) \tag{41}$$

for some $s \in \mathcal{S}$. This contradicts $\tilde{\pi}^*$ being optimal in the model. $\square$

**Proposition 5.** *There exist environments and model classes for which $\mathcal{M}^\infty(\Pi) \subset \mathcal{M}^\infty(\Pi_{\det})$.*

*Proof.* Since the environment and model only differ when action $R$ is taken from state 2, we only need to consider deterministic policies that make this choice. Note that if action $R$ is taken from state 2, the values in both the model and environment at states 2 and 3 are necessarily 0 and the value of each in state 1 is either $(1 - \gamma)^{-1}$ or 0 depending on the action taken from state 1. This suffices to show that the environment and model have the same values for all deterministic policies.

However, one can see that the model and environment differ for stochastic policies. Take, for instance, a policy for which $\pi(a|s) = 0.5$ for all $a \in \mathcal{A}$, $s \in \mathcal{S}$. The induced Markov reward processes from applying this policy to the environment and model, which share the same reward structure, have different transition dynamics at state 2. It can be easily verified that this results in different values for the environment and model. $\square$

**Proposition 6.** *For any $\pi \in \Pi$, $v \in \mathbb{V}$ and $k, n \in \mathbb{Z}^+$, we have that*

$$\|v_\pi - \tilde{\mathcal{T}}_\pi^k v_\pi\|_\infty \leq (\gamma^k + \gamma^n) \underbrace{\|v_\pi - v\|_\infty}_{\epsilon_v} + \underbrace{\|\mathcal{T}_\pi^n v - \tilde{\mathcal{T}}_\pi^k v\|_\infty}_{\epsilon_{ve}}. \tag{10}$$

*Proof.* We begin by considering the left-hand side

$$\begin{aligned}
\|v_\pi - \tilde{\mathcal{T}}_\pi^k v_\pi\|_\infty &= \|v_\pi - \tilde{\mathcal{T}}_\pi^k v + \tilde{\mathcal{T}}_\pi^k v - \tilde{\mathcal{T}}_\pi^k v_\pi\|_\infty \\
&\leq \|v_\pi - \tilde{\mathcal{T}}_\pi^k v\|_\infty + \|\tilde{\mathcal{T}}_\pi^k v - \tilde{\mathcal{T}}_\pi^k v_\pi\|_\infty \\
&= \|v_\pi - \mathcal{T}_\pi^n v + \mathcal{T}_\pi^n v - \tilde{\mathcal{T}}_\pi^k v\|_\infty + \|\tilde{\mathcal{T}}_\pi^k v - \tilde{\mathcal{T}}_\pi^k v_\pi\|_\infty \\
&\leq \|v_\pi - \mathcal{T}_\pi^n v\|_\infty + \|\mathcal{T}_\pi^n v - \tilde{\mathcal{T}}_\pi^k v\|_\infty + \|\tilde{\mathcal{T}}_\pi^k v - \tilde{\mathcal{T}}_\pi^k v_\pi\|_\infty \\
&\leq \gamma^n \|v_\pi - v\|_\infty + \|\mathcal{T}_\pi^n v - \tilde{\mathcal{T}}_\pi^k v\|_\infty + \gamma^k \|v_\pi - v\|_\infty \\
&= (\gamma^k + \gamma^n) \|v_\pi - v\|_\infty + \|\mathcal{T}_\pi^n v - \tilde{\mathcal{T}}_\pi^k v\|_\infty
\end{aligned} \tag{42}$$

as needed. $\square$

**Proposition 7.** *For any $\pi \in \Pi$, $v \in \mathbb{V}$ and $k, n \in \mathbb{Z}^+$, assuming $\|v_\pi - v\|_\infty < g \cdot \|v_\pi - v\|_{d_\pi}$ for some $g \geq 0$, we have that:*

$$\|v_\pi - \tilde{\mathcal{T}}_\pi v_\pi\|_{d_\pi} \leq (g \cdot \gamma^k + \gamma^n) \|v_\pi - v\|_{d_\pi} + \|\mathcal{T}_\pi^n v - \tilde{\mathcal{T}}_\pi^k v\|_{d_\pi} \tag{43}$$

*Proof.*

$$\begin{aligned}
\|v_\pi - \tilde{\mathcal{T}}_\pi^k v_\pi\|_{d_\pi} &= \|v_\pi - \tilde{\mathcal{T}}_\pi^k v + \tilde{\mathcal{T}}_\pi^k v - \tilde{\mathcal{T}}_\pi^k v_\pi\|_{d_\pi} \\
&\leq \|v_\pi - \tilde{\mathcal{T}}_\pi^k v\|_{d_\pi} + \|\tilde{\mathcal{T}}_\pi^k v - \tilde{\mathcal{T}}_\pi^k v_\pi\|_{d_\pi} \\
&= \|v_\pi - \mathcal{T}_\pi^n v + \mathcal{T}_\pi^n v - \tilde{\mathcal{T}}_\pi^k v\|_{d_\pi} + \|\tilde{\mathcal{T}}_\pi^k v - \tilde{\mathcal{T}}_\pi^k v_\pi\|_{d_\pi} \\
&\leq \|v_\pi - \mathcal{T}_\pi^n v\|_{d_\pi} + \|\mathcal{T}_\pi^n v - \tilde{\mathcal{T}}_\pi^k v\|_{d_\pi} + \|\tilde{\mathcal{T}}_\pi^k v - \tilde{\mathcal{T}}_\pi^k v_\pi\|_{d_\pi} \\
&\leq \|v_\pi - \mathcal{T}_\pi^n v\|_{d_\pi} + \|\mathcal{T}_\pi^n v - \tilde{\mathcal{T}}_\pi^k v\|_{d_\pi} + \|\tilde{\mathcal{T}}_\pi^k v - \tilde{\mathcal{T}}_\pi^k v_\pi\|_\infty \\
&\leq \gamma^n \|v_\pi - v\|_{d_\pi} + \|\mathcal{T}_\pi^n v - \tilde{\mathcal{T}}_\pi^k v\|_{d_\pi} + \gamma^k \|v_\pi - v\|_\infty \\
&\leq \gamma^n \|v_\pi - v\|_{d_\pi} + \|\mathcal{T}_\pi^n v - \tilde{\mathcal{T}}_\pi^k v\|_{d_\pi} + g \cdot \gamma^k \|v_\pi - v\|_{d_\pi} \\
&= (g \cdot \gamma^k + \gamma^n)\|v_\pi - v\|_{d_\pi} + \|\mathcal{T}_\pi^n v - \tilde{\mathcal{T}}_\pi^k v\|_{d_\pi}
\end{aligned} \tag{44}$$

$\square$

### A.3 Experimental details - illustrative experiments

#### A.3.1 Code

Code to reproduce our illustrative experiments can be found at `https://github.com/chrisgrimm/proper_value_equivalence`.

#### A.3.2 Computational resources

Illustrative experiments were performed on three machines each with $4$ NVIDIA GeForce GTX 1080 Ti graphics cards.

#### A.3.3 Environment

All illustrative experiments depicted in Figures 3 and 4 were carried out in a stochastic version of the Four Rooms environment (depicted in Figure 7) where $|\mathcal{S}| = 104$ and $\mathcal{A}$ consists of four actions corresponding to an intended movement in each of the cardinal directions. When an agent takes an action, it will move in the intended direction $80\%$ of the time and otherwise move in a random direction. If the agent moves into a wall it will remain in place. When the agent transitions into the upper-right square it receives a reward of $1$, all other transitions yield $0$ reward.

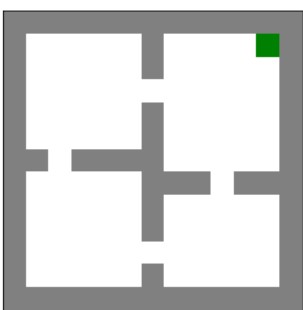

Figure 7: Visualization of the Four Rooms environment.

#### A.3.4 Model representation and initialization

Models are represented tabularly by matrices $\tilde{R} \in \mathbb{R}^{|\mathcal{S}| \times |\mathcal{A}|}$ and $\tilde{P}^a \in \mathbb{R}^{|\mathcal{S}| \times |\mathcal{S}|}$ for $a \in \mathcal{A}$ where $\tilde{R}_{s,a} = \tilde{r}(s,a)$ and $\tilde{P}^a_{s,s'} = \tilde{p}(s'|s,a)$. We generally constrain a matrix to be row-stochastic by parameterizing it with an unconstrained matrix of the same shape and applying a softmax with temperature $1$ to each of its rows. In experiments with model capacity constraints we additionally impose that each $\tilde{P}^a$ has a rank of at most $k$ by representing $\tilde{P}^a = D^a K^a$ where $D^a \in \mathbb{R}^{|\mathcal{S}| \times k}$, $K^a \in \mathbb{R}^{k \times |\mathcal{S}|}$ and both $D^a$ and $K^a$ are constrained to be row-stochastic (note that the product of row-stochastic matrices is itself row-stochastic). In this setting the parameters of the capacity-constrained transition dynamics are the unconstrained matrices parameterizing $D^a$ and $K^a$.

Models are initialized by randomly sampling the entries of $\tilde{R}$ according to $U(-1, 1)$ and the entries of the matrices parameterizing the transition dynamics according to $U(-5, 5)$, where $U(l, u)$ denotes a uniform distribution over the interval $(l, u)$.

In all illustrative experiments we train our models using the Adam optimizer with default hyperparameters ($\beta_1 = 0.99$, $\beta_2 = 0.999$, $\epsilon = $1e-8).

#### A.3.5 Model space experiments

In Figure 3 we illustrate the properties of spaces of models trained to be in $\mathcal{M}^k(\Pi, \mathbb{V})$ for $k \in \{1, 30, 40, 50, 60\}$ and in $\mathcal{M}^\infty(\Pi)$. To train each of these models we construct a set of policies

and functions $\mathcal{D} = \{(\pi_i, v_i)\}_{i=1}^{100,000}$. Each generated policy $\pi_i$ is, with equal probability, either a uniformly sampled deterministic policy or a stochastic policy for which at each state $s$, $\pi_i(a|s) = f_a / \sum_{a \in \mathcal{A}} f_a$ where $f_a \sim U(0,1)$ for each $a \in \mathcal{A}$. Each $v_i$ is sampled such that $v_i(s) \sim U(-1,1)$ for each $s \in \mathcal{S}$. We then sample minibatches $B \sim \mathcal{D}$ with $|B| = 50$ at each iteration and update models to minimize

$$\frac{1}{|B|} \sum_{(\pi,v) \in B} (\tilde{\mathcal{T}}_\pi^k v - \mathcal{T}_\pi^k v)^2 \quad \text{and} \quad \frac{1}{|B|} \sum_{(\pi,v) \in B} (\tilde{\mathcal{T}}_\pi v_\pi - v_\pi)^2 \tag{45}$$

for order-$k$ VE and PVE models respectively.

Each model is updated in this manner for $500,000$ iterations with a learning rate of 1e-3 and a snapshot of the model is stored every 1000 iterations—creating a timeline of the model's progress through training. For each model class, this experiment is repeated with 120 randomly initialized models. To generate the points on the scatter plots depicted in Figure 3, we iterate through the snapshots of these 120 models. At snapshot $t$ (training iteration $1000 \times t$) we collect the snapshots of all the models and convert each model into a 1D vector representation by concatenating the entries from its reward and transition dynamics matrices. We then apply principle component analysis to these vectors, isolating the first two principle components, which we treat as (x, y) coordinates in the scatter plots. For the top row in Figure 3 we color these points according to progress through training: $(t/500)$. On the bottom row, we compute the optimal policy with respect to each point's corresponding model: $\tilde{\pi}^*$ and color the point according to $(\sum_s v_{\tilde{\pi}^*}(s))/(\sum_s v_{\pi^*}(s))$.

We produce the plot of model class diameters in Figure 3 by taking the scatter-plot points corresponding to the final snapshot ($t = 500$) of models for each $k$, randomly grouping them into 4 sets of 30 points and computing the diameters of each set. We then use these 4 diameters to produce error bars.

### A.3.6 Individual model visualization

To generate the visualization of the dynamics of individual models displayed in Figure 4b, we randomly select a single PVE model trained in our model space experiments. We then collect 5000 length 30 trajectories starting from the bottom left state. The paths of these trajectories are then overlaid on top of a visualization of the environment and colored according to time along the trajectory $(t/30)$. This procedure is repeated using the environment in Figure 4a.

### A.3.7 Model capacity experiment

We compare the effect of capacity constraints on learning models in $\mathcal{M}^\infty(\mathbb{\Pi})$ and $\mathcal{M}^\infty(\mathbb{\Pi}_{\text{det}})$ respectively by restricting the rank of the learned model's transition dynamics (as in A.3.4). We restrict the ranks of model transition dynamics to be at most $k$ for $k \in \{20, 30, 40, 50, 60, 70, 80, 90, 100, 104\}$. To train each model we collect a set of 1000 policies by beginning with a random policy and repeatedly running the policy iteration algorithm in the environment, starting with a randomly initialized policy and stopping when the optimal policy is reached. The sequence of improved policies resulting from this process is stored. Whenever the algorithm terminates, a new random policy is generated and the process is repeated until 1000 policies have been stored. To increase the number of distinct policies generated by this process, at each step of policy iteration, we select, uniformly at random, 10% of states and update the policy at only these states. We then further boost the breadth of our collected policies and specialize them to $\mathbb{\Pi}$ and $\mathbb{\Pi}_{\text{det}}$ by adding stochastic or deterministic "noise."

Precisely, when training a model to be in $\mathcal{M}^\infty(\mathbb{\Pi})$ we iterate over each of the 1000 policies generated by our policy iteration procedure and generate an additional 100 policies. Each additional policy is generated by selecting, uniformly at random, 10% of the original policy's states and replacing the its distribution at these states with a uniform distribution over actions.

When training a model to be in $\mathcal{M}^\infty(\mathbb{\Pi}_{\text{det}})$ the same procedure is repeated but the original policy's distributions, at the selected states, are replaced by randomly generated deterministic distributions.

In either case, this produces $100,000$ policies which are evaluated in the environment. Together this forms a set of policies and corresponding value functions: $\mathcal{D} = \{(\pi_i, v_i)\}_{i=1}^{100,000}$ which can be used construct mini-batch PVE losses as described in (45). Models are trained according to these losses for $1,000,000$ iterations with a learning rate of 5e-4. The errorbars around the environment value of the models' optimal policies at the end of training are reported across 10 seeds.

### A.4 MuZero experiment

**Atari.** We follow the Atari configuration used in Schrittwieser et al. [29], summarised in Table 1.

Table 1: Atari hyperparameters.

| PARAMETER | VALUE |
| --- | --- |
| Start no-ops | [0, 30] |
| Terminate on life loss | Yes |
| Action set | Valid actions |
| Max episode length | 30 minutes (108,000 frames) |
| Observation size | $96 \times 96$ |
| Preprocessing | Grayscale |
| Action repetitions | 4 |
| Max-pool over last N action repeat frames | 4 |
| Total environment frames, including skipped frames | 500M |

**MuZero implementation.** Our MuZero implementation largely follows the description given by Schrittwieser et al. [29], but uses a Sebulba distributed architecture as described in Hessel et al. [19], and TD($\lambda$) rather than $n$-step value targets. The hyperparameters are given in Table 2. Our network architecture is the same as used in MuZero [29].

The base MuZero loss is given by

$$\mathcal{L}_t^{\text{base}} = \mathbb{E}_\pi \sum_{k=0}^{K} \left[ \ell^r(r_{t+k}^{\text{target}}, \hat{r}_t^k) + \ell^v(v_{t+k}^{\text{target}}, \hat{v}_t^k) + \ell^\pi(\pi_{t+k}^{\text{target}}, \hat{\pi}_t^k) \right]. \tag{46}$$

The reward loss $\ell^r$ simply regresses the model-predicted rewards to the rewards seen in the environment. To compute the value and policy losses, MuZero performs a Monte-Carlo tree search using the learned model. The policy targets are proportional to the MCTS visitation counts at the root node. The value targets are computed using the MCTS value prediction $\tilde{v}$ and the sequences of rewards. MuZero uses an $n$-step bootstrap return estimate $v_{t+k}^{\text{target}} = \sum_{j=1}^{n} \gamma^{j-1} r_{t+k+j} + \gamma^n \tilde{v}_{t+k+n}$. We use the a TD($\lambda$) return estimate instead.

For our additional loss corresponding to past policies, we periodically store the parameters for the value function and policy (i.e. the network heads that take the model-predicted latent state as input). Then, we compute the same value loss $\ell^v$ for each past value function. To account for the fact that the reward sequence was drawn from the current policy $\pi$ rather than the stored policies, we use V-trace to compute a return estimate for the past policies.

The additional hyperparameters for the buffer of past value heads were tuned on MsPacman, over a buffer size in $\{64, 128, 256\}$ and an update interval in $\{10, 50, 100, 500\}$. Our experiments took roughly 35k TPU-v3 device-hours for both tuning and the full evaluation.

**Full results.** We report the final scores per game in Table 3. The mean scores are across the final 200 episodes in each of three seeds. We also report the standard error of the mean across seeds only. Performing a Wilcoxon signed rank test comparing per-game scores, we find that the version with the additional Past Policies loss has a better final performance with $p = 0.044$.

Table 2: Hyperparameters for our MuZero experiment.

| HYPERPARAMETER | VALUE |
|---|---|
| Batch size | 96 sequences |
| Sequence length | 30 frames |
| Sequence overlap | 10 frames |
| Model unroll length $K$ | 5 |
| Optimiser | Adam |
| Initial learning rate | $1 \times 10^{-4}$ |
| Final learning rate (linear schedule) | 0 |
| Discount | 0.997 |
| Target network update rate | 0.1 |
| Value loss weight | 0.25 |
| Reward loss weight | 1.0 |
| Policy loss weight | 1.0 |
| MCTS number of simulations | 25 |
| $\lambda$ for TD($\lambda$) | 0.8 |
| MCTS Dirichlet prior fraction | 0.3 |
| MCTS Dirichlet prior $\alpha$ | 0.25 |
| Search parameters update rate | 0.1 |
| Value, reward number of bins | 601 |
| Nonlinear value transform | $\text{sgn}(z)(\sqrt{|z|+1}-1)+0.01z$ |
| Value buffer size | 128 |
| Value buffer update interval | 50 |
| Value buffer loss weight | 0.25 |

| Environment | MuZero (our impl.) | | MuZero + Past Policies | |
|---|---|---|---|---|
| alien | 38,698 | ± 2,809 | **52,821** | ± 1,918 |
| amidar | **6,631** | ± 568 | 4,239 | ± 1,550 |
| assault | 35,876 | ± 550 | 35,013 | ± 738 |
| asterix | **674,573** | ± 88,318 | 549,421 | ± 9,280 |
| asteroids | 214,034 | ± 4,719 | **235,543** | ± 14,605 |
| atlantis | 835,445 | ± 92,290 | 845,409 | ± 60,318 |
| bank_heist | 837 | ± 265 | 552 | ± 234 |
| battle_zone | 39,471 | ± 12,658 | **72,183** | ± 11,385 |
| beam_rider | 120,675 | ± 16,588 | 130,129 | ± 14,014 |
| berzerk | 22,449 | ± 3,780 | **35,249** | ± 3,179 |
| bowling | **59** | ± 0 | 47 | ± 7 |
| boxing | 99 | ± 0 | 99 | ± 0 |
| breakout | 504 | ± 165 | **770** | ± 12 |
| centipede | 400,268 | ± 32,821 | **534,432** | ± 38,912 |
| chopper_command | 524,655 | ± 154,540 | 660,503 | ± 27,000 |
| crazy_climber | 189,621 | ± 7,313 | **217,204** | ± 12,764 |
| defender | 322,472 | ± 105,043 | **483,394** | ± 11,589 |
| demon_attack | 131,963 | ± 3,819 | 112,140 | ± 17,739 |
| double_dunk | 3 | ± 4 | -1 | ± 1 |
| enduro | 0 | ± 0 | **132** | ± 86 |
| fishing_derby | -97 | ± 0 | **-52** | ± 29 |
| freeway | 0 | ± 0 | 0 | ± 0 |
| frostbite | 3,439 | ± 1,401 | **8,049** | ± 526 |
| gopher | **121,984** | ± 338 | 120,551 | ± 923 |
| gravitar | 2,807 | ± 123 | **3,927** | ± 54 |
| hero | 7,877 | ± 960 | **9,871** | ± 523 |
| ice_hockey | -6 | ± 4 | -11 | ± 3 |
| jamesbond | **23,475** | ± 1,586 | 13,668 | ± 4,480 |
| kangaroo | 9,659 | ± 2,389 | 10,465 | ± 2,835 |
| krull | 11,259 | ± 173 | 11,295 | ± 108 |
| kung_fu_master | 55,242 | ± 4,267 | **83,705** | ± 6,565 |
| montezuma_revenge | 0 | ± 0 | 0 | ± 0 |
| ms_pacman | 40,263 | ± 387 | **43,700** | ± 1,042 |
| name_this_game | 76,604 | ± 7,107 | **94,974** | ± 9,942 |
| phoenix | 67,119 | ± 9,747 | 49,919 | ± 10,573 |
| pitfall | **-2** | ± 1 | -24 | ± 7 |
| pong | -7 | ± 9 | -6 | ± 9 |
| private_eye | 193 | ± 101 | -6 | ± 228 |
| qbert | 64,732 | ± 8,619 | 70,593 | ± 16,955 |
| riverraid | 27,688 | ± 1,001 | 28,026 | ± 1,823 |
| road_runner | 151,639 | ± 90,186 | **571,829** | ± 106,184 |
| robotank | **53** | ± 2 | 25 | ± 8 |
| seaquest | 27,530 | ± 10,632 | **141,725** | ± 48,000 |
| skiing | **-27,968** | ± 1,346 | -30,062 | ± 248 |
| solaris | 1,544 | ± 140 | 1,501 | ± 193 |
| space_invaders | 3,962 | ± 102 | **5,367** | ± 953 |
| star_gunner | 663,896 | ± 80,698 | 547,226 | ± 126,538 |
| surround | 7 | ± 0 | 6 | ± 1 |
| tennis | -23 | ± 0 | **0** | ± 0 |
| time_pilot | **267,331** | ± 15,256 | 228,282 | ± 10,844 |
| tutankham | 134 | ± 10 | 150 | ± 8 |
| up_n_down | 434,746 | ± 3,905 | 432,240 | ± 4,221 |
| venture | 0 | ± 0 | 0 | ± 0 |
| video_pinball | 376,660 | ± 37,647 | 378,897 | ± 26,486 |
| wizard_of_wor | **79,425** | ± 1,458 | 54,093 | ± 6,294 |
| yars_revenge | 317,803 | ± 62,785 | 423,271 | ± 54,094 |
| zaxxon | 15,752 | ± 231 | 15,790 | ± 196 |

Table 3: Final Atari scores for our deep RL experiments. We report the mean of the final 200 episodes over all three seeds, and the standard error of the mean across seeds.