# OpenReview forum: "Proper Value Equivalence"
_NeurIPS.cc/2021/Conference — NeurIPS 2021 Spotlight_

### Official Review · Reviewer_uAfL · 2021-07-09

**Rating:** 6
**Confidence:** 3

**Summary:**

This paper first extends value equivalency (VE) to k-step value equivalency, where the equivalency holds for k-step application of the Bellman operator. In the limit of k to infinity they then show this becomes ‘proper value equivalence’ (PVE), for which the policy uniquely defines the VE class. They also show that MuZero is a special case of the PVE class, and derive empirical experiments on PVE, also extending MuZero with a new loss.


**Limitations And Societal Impact:**

The authors state societal impact is not applicable, I agree

**Main Review:**

Strong:
– It like the fact that VE starts to get theoretical grounding, and this paper indeed extends the previous ideas of Grimm. It is also nice to see MuZero be put in a theoretical framework.
– The theoretical part seems sound. I am not deeply read up on the theory, and I could therefore not verify all proofs in detail, but they seem correct.
– Apart from the theory, the paper also has experiments, including extensive high-dimensional Atari experiments in which they indeed (slightly) outperform baseline MuZero.

Weak:
– You are quite fast in some of your explanations. For example, Figure 2 is not clear for me, I do not understand what exactly I should see (it needs more text in the caption). In general, I get the feeling you tried to squeeze a lot of information into the paper, and this makes you go pretty fast at several points.
– I miss some more intuition on what happens, especially why the k-step models, in particular the infinity variant (PVE), has such different properties than the one-step VE. You quickly move into the technical terms, but I think it would help if you first gave a more intuitive explanation of what is happening here, why the VE class grows with k, and why this is relevant.
– For your empirical section, I think it would really benefit if you gave an explicit algorithm of your PVE method. It is not entirely clear from the text to me how your overall interaction loop with the environment and training procedure takes place now. I would also prefer to see the exact loss with which you extend MuZero, and how it is embedded in the algorithm (L295).
– I miss a Discussion section, with weaknesses and potential future work.

Smaller:
– Fig 4b: “bottom-right” should be “bottom-left”

Conclusion:
This is a good theoretical paper on an important topic (value equivalency), which has shown state-of-the-art empirical results recently. As far as I can tell, the theoretical results are sound and relevant, although I miss some broader motivation and intuition. The empirical section is good as well, although it would benefit from more explicit algorithms. In general, my biggest issue with the paper is its density: it seems like there was just a bit too little space for both the theoretical and empirical parts (see my comments above), and a discussion is completely omitted. (Maybe the derivation of MuZero being in the PVE class could move to the appendix, and the authors could give additional intuition, explanation and discussion in the main text.) I will nevertheless vote for accept given the positive aspects of the paper.

**Time Spent Reviewing:**

1.5

---

> ### Author Response · Authors · 2021-08-10
> **Reply to Reviewer uAfL**
>
> **It like the fact that VE starts to get theoretical grounding, and this paper indeed extends the previous ideas of Grimm. It is also nice to see MuZero be put in a theoretical framework. – The theoretical part seems sound. I am not deeply read up on the theory, and I could therefore not verify all proofs in detail, but they seem correct. – Apart from the theory, the paper also has experiments, including extensive high-dimensional Atari experiments in which they indeed (slightly) outperform baseline MuZero.**
>
> Thank you! We appreciate the time you put into providing a detailed review of our work.
>
> **You are quite fast in some of your explanations. For example, Figure 2 is not clear for me, I do not understand what exactly I should see (it needs more text in the caption). In general, I get the feeling you tried to squeeze a lot of information into the paper, and this makes you go pretty fast at several points.**
>
> Regarding Figure 2: the purpose of the figure was to highlight the surprising fact that there are models which are PVE with respect to all deterministic policies but not all policies in general (see Proposition 5). This is relevant because both model classes (PVE for all deterministic policies and PVE for all policies in general) are sufficient for planning (see Proposition 4 and Corollary 1). Having this larger class of models that is sufficient for planning is generally favorable for two reasons (1) it requires being PVE with a smaller number of policies and (2) having a larger model class generally means it is easier to learn a model in the class.
>
> The fast pace of the exposition in parts of the text is  a concern that other reviewers raised as well. To address it, we plan on moving some of the more technical points in the derivation in the “MuZero through the lens of value equivalence” section into the Appendix. This will free up space for us to add more exposition, including the discussion above regarding Figure 2.
>
> **For your empirical section, I think it would really benefit if you gave an explicit algorithm of your PVE method. It is not entirely clear from the text to me how your overall interaction loop with the environment and training procedure takes place now. I would also prefer to see the exact loss with which you extend MuZero, and how it is embedded in the algorithm (L295). I miss a Discussion section, with weaknesses and potential future work.**
>
> We will include pseudo-code with a step by step description of the algorithm used in our experiments. We will also provide the exact loss used with MuZero. Several other reviewers also pointed out that we should include a Future work / Limitations section; we plan on including such a section in the revised version of our paper and add some discussion to it (please see the response to the other reviewers)

---

### Official Review · Reviewer_2msG · 2021-07-16

**Rating:** 7
**Confidence:** 4

**Summary:**

This paper introduces a k-th order generalization of the Value Equivalence (VE) principle of Grimm et. al. 2016, generalizing that the candidate value functions are equal under multiple (rather than one) iterations of the Bellman operator for a candidate set of policies. Taking k -> infinity leads to the special case of Proper Value Equivalence (PVE), which only depends on a candidate set of policies. Several structural results and derived, and it is shown that models obtained from PVE (with the set of deterministic policies) have the same optimal policies as the true environment.  A loss function is introduced to learn k-VE models, connections with MuZero are drawn, and experimental simulations are carried out.

**Limitations And Societal Impact:**

See above: “the paper would be greatly strengthened with further theoretical evidence for the quality of PVE models with respect to a smaller (feasible) set of policies, or with more extensive support for the empirical uses of the PVE loss function”.

**Main Review:**

Overall this is a good paper which I recommend for acceptance. There are a couple of issues that I have concerning disconnects between it’s theoretical claims and it’s empirical claims, which I will get into later.

Firstly the paper is well-written and well-organized. The k-th order generalization of VE models is a novel and interesting extension. The theoretical claims are technically sound. The structural properties are interesting and well-investigated (some of them were a bit surprising/counter-intuitive). The result of Corollary 1 is an interesting characterization for the set of models which are sufficient for planning.

However, I am not quite as convinced by the connections with MuZero. The derivation of Sec. 4 establishes that (with some smoothness assumptions), the proposed PVE loss for a singleton policy class is a lower bound to the MuZero loss function. It is not quite clear to me what this is demonstrating. The guarantee of Corollary 1 says that planning is optimal when the policy class is all deterministic policies, and there are no obvious guarantees for the quality of a model that is PVE for a singleton policy class, so this does not seem to give any concrete guarantees for MuZero as far as I can tell.

On the same note, it is not clear to me how useful the loss functions (8) (or (9)) are. The summation over policies seems infeasible, especially since Corollary 1 requires \Pi to be the set of all deterministic policies, which is obviously exponentially large in |S|. Even more, the value function for each of these policies must be estimated. As far as I can tell the full set of policies is never used, and in the MuZero experiments (Figure 5) the loss is minimized for a few policies. This seems to give some empirical advantages, but once again the theory does not indicate why doing this would be helpful.

Overall, the paper would be greatly strengthened with further theoretical evidence for the quality of PVE models with respect to a smaller (feasible) set of policies, or with more extensive support for the empirical uses of the PVE loss function.

Typos: - extra parenthesis in Eq (11)

**Time Spent Reviewing:**

5

---

> ### Author Response · Authors · 2021-08-10
> **Reply to Reviewer 2msG**
>
> **Firstly the paper is well-written and well-organized. The k-th order generalization of VE models is a novel and interesting extension. The theoretical claims are technically sound. The structural properties are interesting and well-investigated (some of them were a bit surprising/counter-intuitive). The result of Corollary 1 is an interesting characterization for the set of models which are sufficient for planning.**
>
> Thank you! We’re glad you appreciated Corollary 1. For another surprising result we would like to direct your attention to Proposition 5 (and Figure 2 for a concrete example) which shows that there are models that are  PVE to all deterministic policies but not PVE to t all policies in general.
>
> **However, I am not quite as convinced by the connections with MuZero. The derivation of Sec. 4 establishes that (with some smoothness assumptions), the proposed PVE loss for a singleton policy class is a lower bound to the MuZero loss function. It is not quite clear to me what this is demonstrating. The guarantee of Corollary 1 says that planning is optimal when the policy class is all deterministic policies, and there are no obvious guarantees for the quality of a model that is PVE for a singleton policy class, so this does not seem to give any concrete guarantees for MuZero as far as I can tell.**
>
> You are correct to point out that Corollary 1 gives no guarantee for PVE model classes of single policies. Our goal in showing that MuZero’s loss is an upper-bound on such a PVE loss is twofold. First, we wanted to make the connection between MuZero and VE more clear. Grimm et al. suggested that VE may help understand the empirical success of MuZero and similar algorithms, but they never provided a detailed account of this connection. We wanted to bridge this gap. Second, in casting MuZero’s loss in terms of VE we set the stage for later modifying it to produce a model that is PVE with respect to a larger set of policies, thus obtaining a model class that is closer to satisfying Corollary 1.
>
> **On the same note, it is not clear to me how useful the loss functions (8) (or (9)) are. The summation over policies seems infeasible, especially since Corollary 1 requires \Pi to be the set of all deterministic policies, which is obviously exponentially large in |S|. Even more, the value function for each of these policies must be estimated. As far as I can tell the full set of policies is never used, and in the MuZero experiments (Figure 5) the loss is minimized for a few policies. This seems to give some empirical advantages, but once again the theory does not indicate why doing this would be helpful.**
>
>
> The inclusion of loss functions described in (8) and (9) is meant to be a general-purpose “recipe” for finding order-k VE / PVE models when the number of functions and policies is finite. We agree that Corollary 1 requires Pi to be a specific (and prohibitively large) set of policies, which we do not use in our experiments for practical reasons. At present, we do not have theoretical results about the performance of PVE models with respect to smaller sets of policies, though it seems likely that performance should improve as we add more policies to a given set. We will add this point to the aforementioned Future Work / Limitations section that we plan on adding in the revised paper.
>
> **Typos**
>
> Thank you for pointing it out!

---

### Official Review · Reviewer_kSxE · 2021-07-16

**Rating:** 7
**Confidence:** 4

**Summary:**

The paper proposes an extension to the value equivalence (VE) principle in MBRL called Proper VE (PVE).
The original VE states that a model is equivalent to the true model if the induced Bellman operators coincide for a set of policies and value functions.
PVE is motivated by the observation that a repeated application of the Bellman operator will eventually converge to a value function of a given policy for a chosen model.
PVE defines a model to be equivalent to the true if, for a set of policies, it results in the same value functions as in the original MDP.
While PVE assumes an infinite Bellman operator application (called order-∞), the paper studies the relation between non-asymptotic versions of PVE.
For example, the authors prove that order-k PVE models are also order-K PVE models if k divides K and demonstrate the result empirically.
The paper shows that PVE models are sufficient for planning, i.e. a policy that is optimal for a PVE model is optimal in the true MDP.
Finally, the authors propose a PVE-inspired loss for model learning and show that MuZero optimizes an upper bound on the loss corresponding to a behavior policy.
Inspired by the observation, the authors extend to the MuZero loss with additional terms corresponding to previous policies and show an increase in the median score on the Atari 57 benchmark.


**Limitations And Societal Impact:**

The answer to the question about the limitations is Yes but the reviewer did not find any relevant discussion.
While the main contribution of the paper is not an algorithm, including a discussion about the limitations would still strengthen the paper.

Some suggestions for improvement are provided in the main review.


**Main Review:**

Originality:

The overall originality of the paper is reasonable.
On one hand, PVE is very closely related to the special case of VE when the chosen set of (arbitrary) value functions coincide with the value functions of the given policies (eq 7).
On the other hand, the paper deepens the understanding of the (P)VE principle by proving a number of propositions.
Prop 1 and Prop 2 provide two different characterizations of PVE models.
Prop 3 shows that PVE models are preferable when an MDP can be factored into a part with the information relevant for planning and an irrelevant part.
Prop 6 shows that the PVE-inspired loss can be upper bounded by a sum of a value approximation error and a difference between results of the application of the model and the true Bellman operators.


Quality:

The overall quality of the paper is high.
In addition to establishing the understanding of the sets of PVE models, the authors visualize the trajectories from the PVE models on the (stochastic) four rooms domain.
Interestingly, PVE models admit transitions that are not allowed by the environment -- could you think about a way to address this phenomenon?


Clarity:

Overall, the paper is well-written.
The main text contains a self-sufficient exposition of results and the appendix contains the details about the experiments and proofs.
The reviewer felt that the pace in some parts of the text (e.g. around Prop 5 and Figure 2) was a bit too high and thus required extra time to understand.
Perhaps moving eq 13 - 18 to the appendix would be a better use of space (e.g. for discussing the limitations of learning value-equivalent models).

Typos:
- L543 \tilde{T}^k_π v is written twice
- L547 “non-empty. and produce”


Significance:

The overall significance of the paper is fair.
The paper provides a foundational block for learning decision-aware models which is an important research direction in RL.
Providing scores for all 57 Atari games would increase the significance of the paper because a scalar can sometimes be insufficient to make conclusions about the performance on 57 games.


Recommendation:

The reviewer leans towards recommending to accept the paper.
Addressing the outlined concerns could improve the quality of the paper and increase the score.


**Time Spent Reviewing:**

5

---

> ### Author Response · Authors · 2021-08-10
> **Reply to Reviewer kSxE**
>
> **The overall quality of the paper is high.**
>
> Thank you! We appreciate the effort you put into your review and plan on using it to improve our paper’s quality even further.
>
> **In addition to establishing the understanding of the sets of PVE models, the authors visualize the trajectories from the PVE models on the (stochastic) four rooms domain. Interestingly, PVE models admit transitions that are not allowed by the environment -- could you think about a way to address this phenomenon?**
>
> Indeed, PVE models admit transitions that are not allowed in the environment. This is, in fact, a desirable property! We show in Proposition 4 there are PVE model classes for which planning with any model in such a class will result in the optimal policy in the actual environment. When these classes contain many models (as in the setting described by Proposition 3, where there is irrelevant information in the environment state), some of these models will be different from the environment yet still be optimal for planning in the environment. One way in which such models may differ from the environment is precisely in the transitions that are allowed or not.
>
> **Overall, the paper is well-written. The main text contains a self-sufficient exposition of results and the appendix contains the details about the experiments and proofs. The reviewer felt that the pace in some parts of the text (e.g. around Prop 5 and Figure 2) was a bit too high and thus required extra time to understand. Perhaps moving eq 13 - 18 to the appendix would be a better use of space (e.g. for discussing the limitations of learning value-equivalent models).**
>
> We agree that the pace of our exposition is too brisk at times. This has been pointed out by other reviewers as well. We plan on taking your advice to move some of the more technical aspects of the derivation in the “MuZero through the lens of value equivalence” section to the appendix to free up space. We can then use this added space to add more exposition in the paper as well as adding a future works / limitations section. We plan on paying particular attention to the exposition around Proposition 5.
>
> **Typos**
>
> Thanks for pointing these out, we’ll fix them in our revised version of the paper.
>
> **The overall significance of the paper is fair. The paper provides a foundational block for learning decision-aware models which is an important research direction in RL. Providing scores for all 57 Atari games would increase the significance of the paper because a scalar can sometimes be insufficient to make conclusions about the performance on 57 games.**
>
> We will include a table with the performance of our modification to MuZero on individual Atari games in order to provide a more comprehensive picture of its performance.
>
> **The answer to the question about the limitations is Yes but the reviewer did not find any relevant discussion. While the main contribution of the paper is not an algorithm, including a discussion about the limitations would still strengthen the paper.**
>
> We do briefly discuss some limitations of the proposed method (e.g., line 292 where we mention the infeasibility of enforcing PVE with respect to all deterministic policies in practice and line 330 where we state that our work assumes that the environment state is provided); however, you are correct in that we should have a more extensive and centralized discussion on the topic. As we mentioned above, we plan on including a limitations / future work section with some of the space freed up by moving the derivation from our “MuZero through the lens of value equivalence” section to the appendix. Some possible directions for future work that we plan on including in that section are (1) the case where value equivalence is only approximately enforced, (2) the performance of PVE models when the policy set is small and (3) connections between our approach and representation learning.

---

> > ### Comment · Reviewer_kSxE · 2021-08-17
> > **After-rebuttal response**
> >
> > Dear Authors,
> > Thank you for the reply.
> > I believe you have addressed most of my and other reviewers' concerns.
> > I keep with my score and recommend accepting the paper.

---

### Official Review · Reviewer_bRZz · 2021-07-19

**Rating:** 8
**Confidence:** 3

**Summary:**

The paper "Proper Value Equivalence" addresses the question of how models for reinforcement learning can be obtained which are sufficient for planning without modelling irrelevant details of the environments. The authors build upon previous work by Grimm et al. and expand the notion of value equivalence to only require value functions actually obtainable by policies in the environment. They show several illustrative examples and proofs of why this notion of proper value equivalence alleviates some problems with previous work, such as not collapsing the space of admissible models down to the optimal model. This is very important in many cases where the true model might not be recoverable due to irreducible approximation errors or missing data.

**Limitations And Societal Impact:**

Limitations of the proposed approach are not discussed, one such a potential limitation was pointed out above.
Since the paper addresses foundational questions of reinforcement learning, a discussion of societal impact seems out of scope.

**Main Review:**

Praise for the paper:

The paper can be seen as a direct successor to Grimm et al. "The value equivalence principle for model-based reinforcement learning". This is not a criticism, the paper builds substantial novel results on this previous work and greatly reduces the gap between the principle outlined by Grimm et al. and practical algorithms. It addresses a clear limitation of the previous work, in which the set of admissible models collapses to the true model, which might not be learnable in practice. PVE shows that more models than just the true model are sufficient for planning and outlines the necessary requirements for these models, which is that they match in value function for all policies in the environments. The propositions and proofs are presented in a manner that is clear to the reader and they are easy to follow.

The paper concludes with a short experimental section in which the authors first show how the successful MuZero algorithm can be understood and modified using proper value equivalence. This presents a very interesting addition to the field since explanations for empirically well performing algorithms are often missing or neglected in the literature.

Criticism of the paper:

Missing from the paper are discussions about the setting in which the Bellman error is not applied directly but minimized by some approximate approach such as Squared Bellman error minimization. Such schemes make up the majority of current literature and addressing the interplay between approximate value iteration and PVE are of significant interest.

The clarity of the paper could be slightly improved by making clear that the resulting $\infty$ Bellman operators are equivalent to the policy value functions under the respective policies. I think the distinction between Grimm et al. general notion of "functions" and the actual value functions of considered policies here could be pointed out more directly.

The paper does not discuss the relationship between proper value equivalence and bisimulation theorems (or related approaches), a topic which has recently become very popular in the field. At least a part of the related work should be used to discuss related work in that field and similarities and differences to the presented approach. The reference to Grimm is a bit unsatisfactory, since that paper also does not provide a real comparison, remaining fairly vague on the topic.

Finally, some few sections of the paper seem unfinished. At at least two places (line 81 and line 134) the authors have added comments in parentheses which feel like they should be expanded into full sentences or footnotes.

Conclusion:

Overall the paper provides novel and interesting insights, is well written and easy to read despite mostly discussing theoretical results. In my eyes, the paper should be accepted to NeurIPS.


Checklist:

Originality: The stated results are novel.
Quality: The proofs seem correct and all theoretical assumptions are listed.
Clarity: The writing style is clear and easy to follow.
Significance: The results build on previous work and expand the understanding of the field of model-based RL. Furthermore they offer a way to gain insight into a SOTA applied RL algorithm.

**Time Spent Reviewing:**

6

---

> ### Author Response · Authors · 2021-08-10
> **Reply to Reviewer bRZz**
>
> Firstly, thank you for taking the time to write your review. We have tried our best to develop and present the concept of proper value equivalence and your feedback helps us make our paper even better.
>
> **Missing from the paper are discussions about the setting in which the Bellman error is not applied directly but minimized by some approximate approach such as Squared Bellman error minimization. Such schemes make up the majority of current literature and addressing the interplay between approximate value iteration and PVE are of significant interest.**
>
> Indeed, in our work we did not address the setting where value equivalence is approximately enforced and we believe this is an exciting avenue for future research. Other reviewers pointed out that a Future Work / Limitations section would benefit the paper. We will add such a section and discuss the potential for approximate value equivalence in it.
>
> **The clarity of the paper could be slightly improved by making clear that the resulting  Bellman operators are equivalent to the policy value functions under the respective policies. I think the distinction between Grimm et al. general notion of "functions" and the actual value functions of considered policies here could be pointed out more directly.**
>
> Yes. The distinction between general functions and value functions is important for understanding the results in our work and we see how a reader could miss this point. We will add more exposition in Section 3 to draw attention to this distinction.
>
> **The paper does not discuss the relationship between proper value equivalence and bisimulation theorems (or related approaches), a topic which has recently become very popular in the field. At least a part of the related work should be used to discuss related work in that field and similarities and differences to the presented approach. The reference to Grimm is a bit unsatisfactory, since that paper also does not provide a real comparison, remaining fairly vague on the topic.**
>
> In our present rendition of the related works section, we referenced some works that involve bisimulation under the banner of “representation learning.” However, you are right: ideas from the theory of bisimulation have been used for model learning in several recent works (such as https://arxiv.org/abs/2006.10742) which merits a separate discussion of them and how they relate to proper value equivalence. We will add this discussion in our related works section.
>
> **Finally, some few sections of the paper seem unfinished. At at least two places (line 81 and line 134) the authors have added comments in parentheses which feel like they should be expanded into full sentences or footnotes.**
>
> We plan on moving some of the explicit derivations of our “MuZero through the lens of value equivalence” section into the appendix to free up space which we will use to expand the parentheticals about VAML and Grimm et al’s properties of value equivalence.

---

> > ### Comment · Reviewer_bRZz · 2021-08-17
> > **Reviewer comment**
> >
> > Dear authors, thanks for engaging so actively with our feedback. I think you covered all the major points raised by me and the other reviewers well and I have no further comments or requests. The rating remains as it is, I believe that this paper should clearly be accepted to NeurIPS.

---

### Decision · Program_Chairs · 2021-09-27

**Decision:**

Accept (Spotlight)

**Comment:**

All reviewers are positive about this work, and they believe the paper provides novel insights to the value equivalency principle. Some issues are brought up in the reviews, most of which are satisfactorily answered in the authors' responses (we had private discussions). Please consult their reviews for the details. Some of them are:

- The paper is sometimes too rushed in its explanations.
Perhaps the authors can remove some of the derivations (e.g., Eqs. 13-18) to an appendix, so that they can expand their discussions elsewhere.
- Some reviewers found the detail insufficient for reproducibility.
- More clear discussion on the relation between bisimulation and value equivalence can be helpful.

The revisions required to improve the paper are minor enough that another round of reviews is not required. Therefore, I recommend the *acceptance* of this paper.

===

In addition to these comments from reviewers, I have some questions and comment that I would appreciate if the authors answer them in their revisions. These are not critical, so they are inconsequential to the decision.

- The loss function in Eq. (8) and (9) have summation over policies (and value functions).
But (P)VE requires the exact match for all policies (and values, for VE).
Requiring an exact match suggests that we need to have a loss that encourages the error to be small *uniformly* over values and policies. This means that we may need to consider the supremum over $v$ and $\pi$, instead of summation over them.

The summation can be just too relaxed to impose the required equivalency. This might be more of an issue when the value and policy space is very large, e.g., infinite number of elements. In that case, a large error in a small (say, zero measure) subset of the value/policy space does not affect the loss at all, but it violates the value equivalency.

If we agree that this is the right/better way to write the loss function, then the losses (8) and (9) might be written as

(8') $\sup_{\pi \in \Pi} \sup_{v \in V} || T_\pi^k v - \tilde{T}_\pi^k v||$

and

(9') $\sup_{\pi \in \Pi} || v_\pi - \tilde{T}_\pi^k v||$.

These show some similarities with VAML. VAML uses the Bellman optimality operator, but we can easily have a version for Bellman operator for a policy $\pi$. In that case, with k = 1, the inner optimization $\sup_{v \in V} || T_\pi^k v - \tilde{T}_\pi^k v||$ would be the original VAML's, and for k > 1, it would be a multi-step extension of VAML (which was not introduced in that paper though).

(2) Is there any typo in Proposition 1(ii)? Currently, we have $\mathbb{M}^k$ there, but shouldn't it be $\mathcal{M}^k$? The same comment for Proposition 3.